# Relationship Between Body Composition and Biomarkers in Adult Females with Breast Cancer: 1-Year Follow-Up Prospective Study

**DOI:** 10.3390/nu17152487

**Published:** 2025-07-30

**Authors:** Angélica Larrad-Sáinz, María Gemma Hernández Núñez, Ana Barabash Bustelo, Inés Gil Prados, Johanna Valerio, José Luis Espadas Gil, María Eugenia Olivares Crespo, María Herrera de la Muela, Blanca Bernaldo Madrid, Irene Serrano García, Ignacio Cristóbal García, Miguel Ángel Rubio-Herrera, Alfonso Luis Calle-Pascual, Juana María Brenes Sánchez, Pilar Matía-Martín

**Affiliations:** 1Endocrinology and Nutrition Department, Hospital Clínico San Carlos, Instituto de Investigación Sanitaria del Hospital Clínico San Carlos (IdISSC), 28040 Madrid, Spain; angelica.larrad@salud.madrid.org (A.L.-S.);; 2Medicin II Department, Facultad de Medicina, Universidad Complutense de Madrid, 28040 Madrid, Spain; 3Centro de Investigación Biomédica en Red de Diabetes y Enfermedades Metabólicas Asociadas (CIBERDEM), 28029 Madrid, Spain; 4Obstetrics and Gynecology Breast Cancer Unit, Women Health Institute, Hospital Clínico San Carlos, 28040 Madrid, Spain; 5Psichooncology, Breast Cancer Unit, Women Health Institute, Hospital Clínico San Carlos, 28040 Madrid, Spain; 6Faculty of Psychology, Universidad Complutense de Madrid, 28040 Madrid, Spain; 7Unidad de Apoyo a la Investigación, Instituto de Investigación Sanitaria del Hospital Clínico San Carlos (IdISSC), 28040 Madrid, Spain; 8Department of Public and Maternal and Child Health, Facultad de Medicina, Universidad Complutense de Madrid, 28040 Madrid, Spain

**Keywords:** breast cancer, body composition, sarcopenic obesity, ultrasonography, bioimpedance

## Abstract

Background/Objectives: After diagnosis, it is common for women with breast cancer to gain weight, which is associated with worse clinical outcomes. However, traditional measures such as body weight, BMI, and waist circumference do not detect key changes in body composition, such as fat redistribution or muscle loss. The objective of this exploratory study was to assess the evolution of body composition and muscle strength after one year of treatment, and their relationship with metabolic biomarkers. Methods: Prospective observational study in newly diagnosed breast cancer patients. Body composition was assessed using bioelectrical impedance analysis (BIA) and ultrasound (US); muscle strength was measured by handgrip dynamometry. Biomarkers analyzed included glucose, insulin, Homeostatic Model Assessment of Insulin Resistance (HOMA-IR), glycosylated hemoglobin (HbA1c), total cholesterol (and its fractions), triglycerides, C-reactive protein (CRP), 6-interleukin (IL-6), vitamin D, myostatin, and fibroblast growth factor 21 (FGF-21). Results: Sixty-one women (mean age 58 years) were included. After one year, fat mass and related parameters significantly increased, while skeletal muscle mass and muscle strength decreased. Sarcopenic obesity prevalence rose from 1.16% to 4.9%. No significant changes were found in biomarkers, but positive correlations were observed between fat parameters and insulin, HOMA-IR, and triglycerides, and negative correlations with HDL-cholesterol. Conclusions: BIA and US can detect unfavorable changes in body composition that are not reflected in conventional measurements. At one year post-diagnosis, women showed increased fat accumulation, muscle loss, and reduced strength, even without significant metabolic biomarker changes. Further research is warranted to elucidate the long-term clinical implications of these findings and the external validity in larger cohorts.

## 1. Introduction

Breast cancer is the most common malignant neoplasm among women worldwide, with increasing incidence, especially in younger populations, despite a progressive decline in mortality [1,2,3]. Treatments have improved survival but often leave long-term health consequences, including cardiovascular disease, diabetes, and altered body composition [4].

Obesity is a well-established risk factor for breast cancer development and worsens clinical outcomes post-diagnosis [5,6]. These effects may be mediated by an altered metabolic profile, characterized by dyslipidaemia and insulin resistance, which can result from changes in body composition, even independently of body mass index (BMI) [7].

In addition to BMI as a general measure of adiposity, central obesity, assessed by waist circumference (WC), has been associated with an increased risk of all-cause mortality, independently of BMI [8]. In this context, abdominal subcutaneous adipose tissue (ASAT) and its deep component (dASAT) may exert more detrimental effects than visceral adipose tissue (VAT) [9,10], although this association remains inconsistent. dASAT has been linked to elevated levels of proinflammatory cytokines (tumor necrosis factor alpha -TNF-α- and interleukin-6, IL-6-), adipokines (leptin, resistin), and increased oxidative stress [11,12], which may promote processes such as angiogenesis, tumor invasion, and the expansion of cancer stem cells [13].

Skeletal muscle tissue has also emerged as a relevant prognostic factor in oncology. Low skeletal muscle mass (SMM) is prevalent in breast cancer and is associated with worse survival and treatment-related outcomes [14,15]. When SMM loss is combined with excess adiposity and reduced muscle strength, the condition is defined as sarcopenic obesity (SO), a clinical entity with more severe outcomes than obesity or sarcopenia alone [16]. However, most studies define sarcopenia solely based on muscle mass, without considering muscle strength, and therefore do not align with the consensus definition proposed by the European Society for Clinical Nutrition and Metabolism (ESPEN) and the European Association for the Study of Obesity (EASO) [17].

To better understand the potential mechanisms underlying changes in body composition, various biomarkers are currently being investigated. Fibroblast growth factor 21 (FGF-21) is a key regulator of energy homeostasis, known to improve glucose and lipid metabolism; its levels increase in response to metabolic stress [18]. In breast cancer patients, FGF-21 has been proposed as a potential biomarker for early diagnosis and prognosis. In this context, high FGF-21 levels at the time of breast cancer diagnosis have been associated with reduced survival, and a decrease in its serum concentration has been reported after one year of endocrine therapy [19,20].

Another emerging biomarker of growing interest is myostatin, or growth differentiation factor 8 (GDF8), which is primarily recognized as a negative regulator of muscle mass [21]. In relation to breast cancer, higher GDF8 expression has been associated with improved survival [22].

Accurate assessment of body composition is essential for identifying these changes. Techniques such as dual-energy X-ray absorptiometry (DXA) and computed tomography (CT) provide highly precise measurements; however, they are less accessible and expose patients to ionizing radiation, which limits their routine use. In clinical practice, bioelectrical impedance analysis (BIA) is widely used due to its low cost and ease of application [23]. More recently, ultrasonography (US) has emerged as a promising tool for evaluating both muscle mass and fat distribution, including differentiated measurements of adipose tissue [24,25].

Studies on changes in body composition assessed using techniques such as BIA and US are limited, as is their impact on changes in metabolic biomarkers. The objective of this exploratory study was to describe the evolution of body composition, including changes in fat distribution and muscle strength, after one year of treatment for recently diagnosed breast cancer, and to analyse their relationship with changes in biomarkers associated with glucose and lipid metabolism, inflammation, and muscle biology. The prevalence progression of sarcopenic obesity (SO) following the ESPEN/EASO criteria was also explored.

## 2. Materials and Methods

### 2.1. Study Design

This is a prospective observational study within the framework of the project “*Lifestyle and Predictive Factors of Weight Gain in Women with Breast Cancer: The MAMAVIDA Project.*” The study began in 2019, and recruitment is ongoing. Newly diagnosed breast cancer patients (stages I–III) from the Breast Pathology Unit at Hospital Clínico San Carlos in Madrid were consecutively invited to participate. Participants were referred to the Nutrition Unit prior to the initiation of any treatment to undergo assessments of body composition and muscle strength. Blood samples were collected, processed, and stored the same day for biochemical analysis. This assessment was scheduled to be repeated annually for up to five years. The women received only general advice on a healthy diet and exercise without any other interventions. The study was approved by the hospital’s ethics committee (CEIm Hospital Clínico San Carlos; ethics code CI 21/009-E) and conducted in accordance with the Declaration of Helsinki.

### 2.2. Study Population

Eligible participants were required to be adults, provide written informed consent, and have no prior history of cancer. Patients with gastrectomy, intestinal resection, pancreatectomy, or any other condition that could compromise their nutritional status were excluded.

As the main project focuses on the evolution of adiposity over time following breast cancer diagnosis, the sample size calculation for this exploratory analysis was based on detecting changes in the most clinically accessible adiposity marker, %FM estimated by BIA. Accordingly, a sample size of 44 paired observations would be required to reliably detect an effect size of δ ≥ 0.5 with a statistical power greater than 0.9, assuming a two-sided test and a maximum Type I error rate of α = 0.05. For US comparisons, the sample size estimate was more conservative due to the lack of established data on minimally clinically relevant changes. In this context, 52 paired observations would be needed to detect an effect size of δ ≥ 0.4 with a power greater than 0.8, also assuming a two-sided test and α = 0.05.

### 2.3. Clinical and Anthropometric Data

Clinical data were collected by physicians from the Breast Pathology Unit. For this study, information was gathered on tumor subtype (in situ, luminal, HER2-positive, and triple-negative) and hormonal status (premenopausal or postmenopausal). Additionally, data on the medical history of type 2 diabetes mellitus (T2DM), dyslipidaemia, hypertension, and tobacco use were recorded.

Anthropometric and body composition assessments were conducted in the Nutrition Unit. All measurements were performed in the morning following a minimum 4-h fast. At the initial visit, height was measured with the patient standing barefoot, feet together, head upright, and facing forward, using a Seca^®^ 220 stadiometer (Seca GmbH & Co. KG, Hamburg, Germany). WC was measured according to the guidelines of the Spanish Society for the Study of Obesity (SEEDO), using a flexible, non-elastic measuring tape placed at the midpoint between the lower margin of the last rib and the upper border of the iliac crest, at the end of a normal expiration. The waist-to-height ratio (WHtR) was subsequently calculated as WC (cm) divided by height (cm). Weight was obtained using the scale integrated into the Seca^®^ mBCA 515 BIA (Seca GmbH & Co. KG, Hamburg, Germany). BMI was calculated using the standard formula: weight (kg) divided by height squared (m^2^).

### 2.4. Body Composition Measurement

Body composition was assessed using the Seca^®^ mBCA 515 BIA (Seca GmbH & Co. KG, Hamburg, Germany), a multifrequency, segmental, 8-electrode device. Measurements were performed with the participant in an upright standing position, in direct contact with the hand and foot electrodes, free of any metallic objects. Patients with medical devices such as pacemakers or electrical stimulators were excluded from BIA assessment. Patients with orthopaedic prostheses were included, but analyses showing atypical or inconsistent results were excluded.

The device provided data on resistance (Rz) and reactance (Xc) at 50 kHz, as well as estimation of fat mass (FM) in kilograms (kg) and percentage (%), visceral fat (VF) in litres (L), fat-free mass (FFM) in kg, phase angle in degree angle (PhA), total body water (TBW) and extracellular water (ECW) in L, and their relative distribution expressed as the ECW/TBW ratio in %. The PhA/BMI and the FM/FFM ratios were also calculated.

SMM was estimated using the Janssen formula:SMM (kg) = ((height -cm-^2^/Rz) × 0.401) + (gender × 3.825) + (age × −0.071) + 5.102,
where gender is coded as 0 for females and 1 for males; this formula has been validated across a wide BMI range (16–48 kg/m^2^) [26].

Appendicular skeletal muscle mass (ASMM) was calculated using the Sergi formula:ASMM (kg) = −3.964 + (0.227 × (height -cm-^2^/Rz)) + (0.095 × weight (kg)) + (1.384 × gender) + (0.064 × Xc),
where gender is coded as 1 for males and 0 for females; this formula has been validated in adult populations [27].

Indexes were calculated by adjusting for height squared:

Fat-Free Mass Index (FFMI = FFM/height^2^), Skeletal Muscle Index (SMMI = SMM/height^2^), and Appendicular Skeletal Muscle Mass Index (ASMMI = ASMM/height^2^).

Relative skeletal muscle mass (%SMM) was calculated as: (SMM/weight (kg)) × 100. An additional ratio was calculated as ASMM adjusted for BMI (ASMM/BMI).

### 2.5. Nutritional Ultrasonography

Nutritional US was performed using a Mindray Z60 ultrasound system (Mindray Medical International Limited, Shenzhen, China; distributed in Europe by Probo Medical Ltd., Nottinghamshire, UK) with a linear transducer operating in the 5–10 MHz range. The measurements were taken by two trained professionals. Assessments were conducted with the patient in the supine position. With the probe placed transversely and without applying pressure, the assessment was made at the distal third between the superior border of the patella and the anterior superior iliac spine. Measurements included rectus femoris (RF) thickness, muscle cross-sectional area (RF-CSA), and thigh SAT.

At the abdominal level, assessments were performed at the midpoint between the xiphoid process and the umbilicus. Measurements included total abdominal subcutaneous adipose tissue (tASAT), superficial abdominal subcutaneous adipose tissue (sASAT), dASAT, and preperitoneal adipose tissue (preperitoneal AT).

Each parameter was measured three times, and the average of the three measurements was used for analysis.

### 2.6. Assessment of Grip Strength

Muscle strength was assessed using the Jamar^®^ Plus digital dynamometer (Performance Health Supply Inc., Nottinghamshire, UK). With the patient seated, the dominant arm positioned at a 90° angle and unsupported, the patient was instructed to squeeze with maximal effort. The dynamometer handle was adjusted according to individual hand size to optimize grip alignment. Three measurements were taken, each separated by a one-minute rest interval, and the mean value was used as the final result. In addition, the handgrip strength-to-skeletal muscle mass ratio (HGS/SMM) was calculated as an indicator of muscle quality [28].

### 2.7. Diagnosis of Sarcopenic Obesity

The diagnosis of SO was based on the ESPEN/EASO guidelines [17]. Patients had to meet the criteria of low HGS < 16 kg along with low relative SMM < 27.6%, and FM > 43%.

### 2.8. Biomarkers

Venous blood samples were collected at two time points: at baseline and after 12 months of follow-up. All samples were drawn in the morning following a minimum 4-h fast using standard venipuncture into serum-separating tubes. The following biomarkers were assessed.

Total cholesterol was quantitatively determined using the enzymatic colorimetric cholesterol assay (CHOD-PAP). Serum HDL cholesterol levels were measured using the enzymatic immunoinhibitory method on an Olympus 5800 analyzer (Beckman-Coulter, Brea, CA, USA). LDL cholesterol was calculated using the Friedewald formula. Serum triglycerides were quantified using an enzymatic colorimetric method employing glycerol phosphate oxidase p-aminophenazone (GPO-PAP).

C-reactive protein (CRP) levels were measured nephelometrically using the Dimension Vista system (Siemens Healthcare Diagnostics, Munich, Germany). Fasting serum insulin (FSI) was measured by chemiluminescent immunoassay on an IMMULITE 2000 Xpi analyzer (Siemens Healthcare Diagnostics, Munich, Germany), with inter-assay precision of 6.3% at a concentration of 11 µIU/mL and 5.91% at 21 µIU/mL.

The homeostasis model assessment of insulin resistance (HOMA-IR) was calculated as glucose (mmol/L) × insulin (µIU/mL)/22.7. Glucose (glucose oxidase method) and glycosylated haemoglobin (HbA1c, %) levels were standardized according to the International Federation of Clinical Chemistry and Laboratory Medicine (IFCC) using high-performance ion-exchange gradient liquid chromatography on a Tosoh G8 analyzer (Tosoh Co., Tokyo, Japan). Inter-assay imprecision for HbA1c at 5.1% showed a standard deviation (SD) of 0.06 and a coefficient of variation (CV) of 1.23%; at 10.39% HbA1c, SD was 0.11 and CV was 1.04%.

For myostatin and FGF-21, after allowing clot formation for 30 min at room temperature, samples were centrifuged at 1700× *g* for 10 min. Serum was aliquoted and stored at −80 °C until analysis. All samples underwent only a single freeze-thaw cycle prior to biochemical determinations. Serum total myostatin and FGF-21 concentrations were measured in duplicate using a competitive ELISA (EIA-5691, DRG Instruments GmbH, Marburg, Germany) and a sandwich ELISA (RD191108200R, BioVendor, Brno, Czech Republic), respectively. Both assays were performed on a fully automated Dynex DSX ELISA processing system (Dynex Technologies, Chantilly, VA, USA) following the manufacturer’s protocols. For myostatin, serum samples were diluted 1:10, pre-incubated with biotinylated tracer, and transferred into anti-myostatin antibody-coated microplates. After a 2-h incubation, plates were washed and incubated with streptavidin-HRP conjugate for 1 h, followed by substrate development (tetramethylbenzidine) and absorbance reading at 450 nm (reference 620 nm). Concentrations were calculated by 4-parameter logistic regression and corrected for dilution. The assay sensitivity was 0.37 ng/mL; intra- and inter-assay coefficients of variation were below 15%.

For FGF-21, serum samples were diluted 1:2 and incubated for 1 h in microplates pre-coated with anti-FGF-21 antibodies, followed by sequential incubations with biotin-labeled detection antibody (1 h) and streptavidin-HRP conjugate (30 min). After substrate reaction, absorbance was read at 450 nm with 630 nm reference. Concentrations were calculated using a 4-parameter logistic curve, adjusted for dilution. The assay detection limit was 7 pg/mL, with intra- and inter-assay coefficients of variation below 5%.

### 2.9. Statistical Analysis

Continuous variables were expressed as means (standard deviation, SD) or medians (interquartile range, IQR). Categorical variables were presented as absolute frequencies and percentages. The distribution of continuous variables was assessed using the Kolmogorov–Smirnov test. Differences in baseline characteristics of the sample by hormonal status were evaluated using the chi-square test or Fisher’s exact test, as applicable. Age was compared with Student’s T test for independent samples.

The comparisons between basal and one-year quantitative parameters were assessed using the paired T-Student test or the Wilcoxon test, depending on their distribution.

Bivariate correlations between body composition changes and biomarkers evolution (absolute, Δ, and relative -%- differences) were evaluated by Spearman’s correlation coefficients after assessing the non-normality of the variables.

To account for confounders in the associations between changes in body composition parameters and biochemical biomarkers, we performed both crude and age-adjusted linear regression analyses using rank-transformed variables that showed statistically significant associations in the correlation analysis. Since the body composition parameters (independent variables) were interrelated and given the limited sample size, only age (rank-transformed) was included as a covariate.

Patients with missing values have not been considered in the analyses involving those variables. All statistical tests were two-tailed, and a *p*-value < 0.05 was considered statistically significant. Analyses were performed using the Statistical Package for the Social Sciences (SPSS), version 25.0 for Windows (IBM Corp., Armonk, NY, USA). A correlation matrix was computed using R (versión 4.5.1) and visualized with the *corrplot* package (version: 0.92).

## 3. Results

### 3.1. Baseline Description

This study involved sixty-one women who underwent breast cancer surgery (73.8% in the menopausal stage; mean age 58 years, range 40–80). Table 1 describes the baseline characteristics of the sample by hormonal staging. The most prevalent tumor subtype was luminal disease, present in 67.2% of patients, corresponding to 39.3% Luminal A and 27.9% Luminal B. According to hormonal status, luminal disease accounted for 75.1% and 64.5%, and HER2+ tumors for 6.3% and 6.6%, in pre- and postmenopausal women, respectively. A triple-negative tumor was only observed in one postmenopausal woman. After surgery, 23.0, 73.8, and 85.3% received chemotherapy, radiation, and hormone therapy, respectively. The proportion of women treated with tamoxifen, letrozole, and anastrozole was different in pre- and postmenopausal patients (*p* = 0.014). Hypertension or dyslipidaemia was observed in 37.7% of patients (25.0% of premenopausal and 42.2% of postmenopausal women). Type 2 diabetes mellitus (T2DM) occurred in 4.9% of cases, with a higher prevalence in premenopausal (6.3%) than in postmenopausal women (4.4%). A total of 86.9% of patients reported never having smoked.

### 3.2. Body Composition and Muscle Strength Evolution: Sarcopenic Obesity Prevalence

The evolution of anthropometry, body composition assessed by BIA, and muscle strength is presented in Table 2. Body weight and BMI increased by 1.4% one year after diagnosis (*p* = 0.021 and 0.014, respectively), as did WC and WHtR, but these differences did not reach statistical significance. Regarding the BIA variables, Rz did not change, whereas Xc and PhA decreased significantly by 3.83% and 2.08%, respectively. ECW/TBW ratio increased by 1.36%. All fat mass-related parameters, except VF, increased significantly (FM (kg) 3.58%, FM (%) 2.20%, FMI 3.62%, and FM/FFM ratio 4.37%). Among muscle-related measures, FFMI and ASMMI did not show significant changes, while SMMI (–1.15%), ASMM/BMI (–1.64%), and %SMM/weight (–2.90%) decreased (*p* < 0.05 for all comparisons). Absolute HGS remained unchanged, while HGS adjusted for SMM decreased by 3.02% (*p* = 0.037).

Concerning the progression of US measurements (Table 3), at the abdominal level, tASAT increased by 7.03% and dASAT by 14.44% (*p* < 0.05), while sASAT showed a smaller increase (1.24%), which was non-statistically significant. Preperitoneal AT also experienced a relevant increase of 15.19% (*p* = 0.031). At the thigh and RF level, all measures changed significantly. SAT increased by 24.16%, RF thickness by 11.05%, and RF-CSA by 14.09%, respectively (*p* < 0.005).

The prevalence of SO increased from 1.16% to 4.9% after one year of follow-up (1 and 3 patients, respectively).

### 3.3. Biomarkers Evolution

Although no significant changes were detected in any of the biomarkers, increases were observed in blood glucose (1.7%) and HOMA-IR (6.6%), as well as a slight increase in insulin levels (0.6%). Regarding the lipid profile, decreases were recorded in all cholesterol fractions: total cholesterol (2.6%), HDL (0.5%), non-HDL (7.0%), and LDL (9.0%). In contrast, triglycerides increased by 7.9%. Additionally, increases were detected in myostatin (1.6%) and FGF-21 (32%) levels (Table 4).

### 3.4. Association Between Body Composition Changes and Biomarkers Evolution

The correlations between changes in metabolic markers and body composition parameters are shown in Figure 1 and detailed in the Appendix A. Briefly, insulin and HOMA-IR were positively associated with FM and negatively with muscle measures, estimated with BIA. The strongest correlations were found between absolute %FM (r = 0.522; *p* = 0.001) and FM/FFM (r = 0.538; *p* = 0.001) changes, and the absolute HOMA-IR difference after one year of follow-up. Regarding the lipid profile, the most notable associations were found with the US measures. Both tASAT and sASAT percentual changes were negatively associated with the relative HDL cholesterol difference at 1 year (r = –0.461; *p* = 0.001 and r = –0.414; *p* = 0.004, respectively) and positively with triglycerides changes (r = 0.377; *p* = 0.008 for absolute differences with tASAT, and r = 0.468; *p* = 0.010 for percentual differences with sASAT). Among the inflammation parameters, CRP was negatively associated with ASMM/BMI (r = −0.350; *p* = 0.012 and r = −0.337; *p* = 0.016 for absolute and relative changes, respectively), while IL-6 correlated with WC (r = −0.325; *p* = 0.020) and sASAT (r = 0.337; *p* = 0.025) when absolute changes were considered. The changes in vitamin D levels were negatively correlated with the evolution of body weight (r = −0.344; *p* = 0.012), BMI (r = −0.311; *p* = 0.024), and preperitoneal AT (r = −0.300; *p* = 0.045), described in absolute terms. Finally, myostatin was negatively associated with weight (r = −0.362; *p* = 0.020), BMI (r = −0.361; *p* = 0.020), and RF thickness (r = −0.329; *p* = 0.047) -absolute differences-. The changes in glucose and FGF-21 levels were not consistently correlated with any body composition parameter evolution.

The crude and age-adjusted linear regression models are presented in the Appendix A. Robust and biologically plausible associations were observed between body composition parameters and key metabolic biomarkers after adjustment for age. Increased FM, as measured by absolute or relative changes in %FM, relative changes in FMI, and absolute changes in FM/FFM ratio, showed strong positive associations with absolute and relative changes in HOMA-IR (adjusted β ranging from 0.400 to 0.487; *p* < 0.025). Increased SAT, as measured by absolute and relative changes in tASAT and relative changes in sASAT demonstrated strong negative associations with absolute and relative changes in HDL-cholesterol (adjusted β ranging from 0.408 to 0.456; *p* < 0.005). In addition, absolute change in sASAT was significantly and positively associated with absolute change in triglycerides (adjusted β 0.405; *p* = 0.005). However, some of the studied associations were no longer significant after adjusting for age.

## 4. Discussion

In this exploratory study, changes in body composition and muscle quality were observed in women within the first year following diagnosis of breast cancer. A general increase in fat mass was noted, particularly in the deeper layers of abdominal adipose tissue, as assessed by US. However, these changes were not mirrored by significant variations in WC. In addition, some muscle tissue parameters, mainly those normalized by weight and BMI, exhibited signs of deterioration in both quantity and quality (HGS/SMM), potentially indicating an early trajectory towards long-term functional decline. Our analyses revealed several robust and statistically significant associations between changes in body composition, classical anthropometric parameters, and biochemical biomarkers, independent of age. Notably, increases in FM—expressed as percentage FM and FMI—were strongly and positively associated with markers of insulin resistance, including HOMA-IR. Conversely, SAT in the thigh and abdominal regions showed consistent inverse relationships with HDL-cholesterol and positive associations with triglycerides, highlighting their impact on lipid metabolism. Weight gain following diagnosis is a common alteration and has been consistently associated with adverse clinical outcomes. These include not only an increased risk of mortality and tumor recurrence [6], but also an increased risk of cardiovascular disease [29] and metabolic dysfunction [30,31,32]. Although anthropometric indicators such as body weight, BMI, and WC are widely employed to assess nutritional status and cardiometabolic risk, they fail to accurately capture key alterations in body composition, such as fat redistribution and skeletal muscle loss [33].

In our cohort, modest but statistically significant increases were observed in weight (1 kg; +1.4%) and BMI (0.4 kg/m^2^; +1.4%), with no relevant changes in WC (0.6 cm; +0.7%) one year after diagnosis. These findings are consistent with the review by Pedersen et al., which reported weight gains ranging from 0.9 to 3.9 kg (0.39–5.9%) over the same period [34]. Despite maintaining a BMI within normal ranges, participants presented with elevated FM percentages and WC values nearing risk thresholds, which highlights the limitations of BMI as an independent marker of adiposity and cardiometabolic risk [35].

It is of interest that, during the early post-diagnostic period, fluctuations in body weight may be partially attributable to increases in TBW [36]. In this context, BIA represents a valuable tool for detecting such shifts. In our study, we observed a significant increase in ECW/TBW, from 46% to 47%. This phenomenon is associated with an increase in fat mass [37] as well as metabolic stress induced by oncological treatments [38]. Silva et al. reported comparable results, documenting an elevation in the ECW/TBW from 45% to 48% [39]. Other studies have described transient elevations in TBW during the initial months following diagnosis, followed by a subsequent return to baseline levels [34,40].

This increase in TBW represents a limitation of BIA, as it may lead to an underestimation of FM and an overestimation of FFM in these patients [41]. In our cohort, FFMI remained unchanged at one-year post-diagnosis; however, a >2% increase in FM percentage was observed, potentially indicating the early onset of an SO phenotype. This trend in body composition is consistent with other studies reporting similar increases in FM without significant changes in FFM [34,42,43]. It is important to consider that, although part of the initial weight gain may be attributed to an increase in ECW, only 10% of women manage to return to their pre-diagnosis weight, suggesting that changes in body composition may persist in the long term [44].

Given this limitation, PhA, a parameter derived from Rz and Xc, has gained interest as a sensitive marker of cellular integrity [45]. In breast cancer survivors, a progressive decline in this parameter has been observed, which is associated with poorer prognosis [46,47]. We observed a significant decline in PhA during the first year of follow-up, accompanied by a decrease in Xc, which reflects cellular damage due to oxidative stress [39]. Since oxidative stress disrupts cell membranes and promotes catabolic processes, a low PhA has been proposed as an indirect marker of this imbalance [48]. Moreover, PhA has also been proposed as an alternative indicator of muscle quality [49]. Nonetheless, further research is warranted to validate this association and to establish its long-term clinical relevance.

Regarding fat distribution, significant increases were observed in subcutaneous adipose tissue both in the thigh (+24.16%) and in the abdominal region (+7.03% tASAT, +1.24% sASAT, and +14.44% dASAT), as well as in preperitoneal AT (+15.16%). The increase in VF, assessed by BIA, was not statistically significant. Although reference values for US assessment are not yet available, studies using CT and DXA have reported similar findings. Specifically, when CT was employed as a tool for body composition analysis, comparable changes were documented, including increases of 7.5% in tASAT and 11.7% in VAT [50]. DXA-based studies have reported significant increases in leg fat (7%), android fat (8.2%), gynoid fat (5.6%), and VAT (18.9%) [51,52]. When considered alongside reports indicating reductions in trunk muscle mass among breast cancer patients [51,52,53], this pattern reflects a redistribution of body composition that is not captured by conventional anthropometric measurements. This evidence supports the utility of US as a promising, non-invasive method for monitoring body composition in this patient population.

Although lean mass tends to decrease [51,52,54], studies have shown that this decline predominantly affects the trunk, with lesser involvement of the extremities [51,52,53]. Differences in segmental muscle loss may explain the stable ASMMI values observed in our cohort, despite a measurable reduction in SMMI. Nevertheless, several studies have documented the absence of significant alterations in this muscular parameter [50]. This apparent lack of change should not be interpreted as an absence of muscle involvement, but rather as a reflection of a more complex process of skeletal muscle deconditioning [55]. This is particularly relevant in breast cancer patients whose nutritional status is not compromised [56], and in whom, therefore, the loss of muscle mass cannot be attributed to a deficiency of energy or protein substrates, but to mechanisms related to treatment and inactivity. As emphasized by Mallard et al., skeletal muscle deterioration follows a progressive course rather than occurring abruptly [55]. Early stages of muscle alteration, which may precede overt structural atrophy, include interstitial edema, fat infiltration, and mitochondrial dysfunction, factors that may transiently obscure or compensate for reductions in measurable muscle size. This phenomenon has been corroborated by studies employing more sensitive techniques, such as CT and muscle biopsy, which have demonstrated that, even in the absence of apparent muscle loss, reductions in muscle density, fatty infiltration, and decreased muscle fiber CSA can occur, indicating early functional decline [38,57,58,59]. This may help explain why, contrary to expectations, a significant increase in RF thickness was observed in our cohort one year after diagnosis. While these measurements must be interpreted with caution, it is possible that US, like BIA, may be subject to measurement inaccuracies in the presence of altered tissue hydration (60). Although muscle US has the potential to assess fat infiltration and edema through echogenicity analysis, this parameter was not evaluated in our study, which limits the interpretation of these findings.

An indicator of progressive muscle deterioration is the decline in the tissue’s capacity to generate force, reflecting a reduction in muscle quality [60]. In this context, even in the absence of notable absolute changes in muscle strength as assessed by dynamometry, as reported in our study (23 to 22 kg) and in that of Godinho-Mota et al. (22 to 21 kg) [54], a significant reduction in muscle quality, as indicated by the HGS/SMM ratio, was evident during the first year of follow-up, potentially representing an early marker of skeletal muscle deconditioning. In muscle deconditioning, the initial changes are often structural and may not be detectable with conventional assessment methods such as HGS. One such example is myosteatosis, which can precede functional decline. Typically, loss of muscle strength occurs at a later stage, once structural alterations are more established.

The alteration of the balance of body composition was clear when body tissues and muscle mass measurements were adjusted to body size (FM/FFM, %SMM/weight, ASMM/BMI), as these proportions reflect an altered ratio of fat mass to lean mass. The concept of load-capacity, this disproportionate increase in metabolic load (body fat) relative to metabolic capacity (muscle mass) contributes to long-term functional and metabolic decline [61]. Its elevation has been associated with cardiometabolic disturbances and an increased risk of mortality in the general population [62,63,64].

SO has been proposed as a distinct body composition phenotype, defined by the simultaneous presence of excess adiposity, low muscle mass, and reduced muscle strength. To date, only one study has analysed OS in breast cancer survivors based on data similar to the consensus definition proposed by ESPEN/EASO, reporting a prevalence of 16% [65]. This prevalence is considerably higher than that observed in our cohort (1.16–3.49% at baseline and at one-year follow-up, respectively). It should be noted that the referenced study included older patients, which may have contributed to a higher prevalence of sarcopenia. In addition, the authors used different diagnostic criteria (SMM/weight < 30.7%, FM > 31.7% and HGS < 20 kg) than those we applied in our analysis with a more conservative approach (SMM/weight < 27.6%, FM > 43% and HGS < 16 kg). Nevertheless, our findings showed a slight increase in the prevalence of OS at one-year post-diagnosis, which may lend support to the previously discussed observations. However, this must be interpreted with caution given the limited number of women involved. Moreover, current cut-off points for sarcopenic obesity diagnosis may not be appropriate for younger populations, which is one of the main challenges when studying body composition in early adulthood. These types of limitations also highlight the importance of adapting diagnostic criteria to different age groups and population characteristics.

Although sarcopenia has traditionally been regarded as an age-related condition, accumulating evidence suggests that obesity may independently induce a loss of muscle mass and function. This association is attributed to the deleterious effects of metabolic disturbances linked to AT, such as oxidative stress, chronic inflammation, and insulin resistance, which exert a harmful impact on skeletal muscle [17]. Consequently, sarcopenia can occur at any age, although its diagnosis in young populations is difficult because the currently established criteria are designed for older people, which could explain the low prevalence observed in our sample. Even so, the combined impact of obesity and cancer treatments could favour the early onset of this condition, accelerating its development compared to women who have not suffered from the disease [66].

In our cohort, no relevant changes were observed in the analysed biomarkers after one year of follow-up. As noted by Makari-Judson et al., these alterations tend to occur primarily during chemotherapy, followed by a subsequent normalisation, a pattern that is consistent with our findings [30]. It is important to note that studies reporting significant changes typically assess patients immediately after the completion of treatment, rather than during long-term follow-up [43,54,67].

Although the correlations observed between body composition and metabolic biomarkers were not always consistent and, in some cases, contrary to expectations, we were able to identify interesting associations with fat tissue distribution and certain biomarkers. In this context, a detailed analysis of body composition, using techniques such as BIA or US, may allow for earlier detection of combined risk metabolic phenotypes, exposing patients to the development of cardiovascular disease, frailty, and premature aging.

In our study, the increase of sASAT (as in other measures of adiposity) was found to be correlated with decreased HDL-cholesterol levels and increased triglycerides. This pattern suggests an unfavourable metabolic profile which, if sustained over time, may translate into an increased risk of cardiovascular disease [68]. Total adiposity changes, estimated by BIA, showed positive correlations with insulin and HOMA differences after 1 year of follow-up, whereas some muscle mass variables demonstrated inverse associations. Guian et al. reported a relationship between BMI, insulin, and HOMA at both diagnosis and three-year follow-up. However, WC only showed associations at follow-up, with no apparent changes in its measurement [42], suggesting that redistribution of body compartments could be one of the reasons. We could not demonstrate those correlations between anthropometric variables and carbohydrate metabolism in the evolution from the diagnosis.

Regarding inflammatory markers, CRP change showed positive correlations with several measures of adiposity, whereas IL-6 evolution was only associated positively with sASAT, with no relationship with dASAT, as reported in previous studies [11].

As expected, the evolution of the vitamin D levels correlated negatively with adiposity-related parameters, such as BMI, body weight, and preperitoneal AT. In our sample, we could not assess the frequency of supplementation, but its values remained stable after one year. Given the evidence relating vitamin D and breast cancer prognosis [68], strategies to reduce the fatness increase after the diagnosis seem to be prioritized.

Our results show stable levels of FGF-21 and myostatin. About myostatin, we observed a negative correlation with body weight and BMI, which contrasts with existing evidence [69]. However, we have described a negative correlation with RF thickness that aligns myostatin with its role as a negative regulator of muscle growth. In breast cancer, elevated myostatin expression has been associated with better prognosis and overall survival [22], so further research is required to clarify this association and determine its clinical relevance.

The results of both crude and age-adjusted linear regression models revealed several statistically significant associations between changes in body composition, classical anthropometric parameters, and biochemical biomarkers. However, some of these associations were no longer significant after adjusting for age. This attenuation of significance after adjustment suggests that age influences these associations and may partly explain the correlations observed in the crude analyses.

The small sample size is one of the main limitations of our study. From an initial recruitment of 138 participants, we selected the sample that completed the protocol after one year of follow-up, representing 44.2% of the cohort. Baseline characteristics—including age, hormonal status at diagnosis, tumor type, and comorbidities—were comparable between the total sample and this subset. Although such attrition may affect internal and external validity, the strongest changes and associations observed can be considered in future prospective studies, modeled with additional explanatory variables. The Appendix A presents the results of post hoc power analyses for variables (body composition, strength, and biochemical biomarkers) in which changes after one year of follow-up were not statistically significant. These analyses indicate that the non-significant findings may be attributable to insufficient statistical power, which ranged from 0.050 to 0.664. An increased sample size would be required to detect statistically significant differences. Moderate to large correlations (±0.35 and above) generally achieved acceptable power (>0.70), whereas weaker correlations (<0.30) were underpowered. Therefore, the lack of significance in correlations with low rho values in our study (approximately 0.10–0.20) may also be due to insufficient power. In contrast, significant correlations with moderate to large rho values were adequately powered. However, given the multiple comparisons conducted simultaneously in the correlation and regression analyses, the reported significance levels should be interpreted with caution. Applying the Bonferroni correction, *p* values in the range of 0.004 to 0.05 may still carry a risk of Type I error. In Appendix A, the most robust *p* values have been highlighted. Therefore, this sample size has not allowed us to perform valid sub-analyses based on menopausal status, exercise, dietary habits, or the treatment received, which have been identified as risk factors for deleterious changes in body composition [33]. On the other hand, BIA analysis may not be accurate enough to estimate body compartments in patients at the extremes of age or BMI, or with altered hydration status. Nevertheless, the changes observed during follow-up were consistent with previously published studies [43], and although an increase in ECW/TBW was observed, the percentage and absolute values of TBW did not change in a clinically relevant manner (from 28.33 ± 3.82 L to 28.40 ± 3.63 L, and from 45.69 ± 4.92% to 44.75 ± 4.74%). More studies are necessary to assess the external validity of BIA-derived data in breast cancer. Moreover, the US assessment was performed by two different investigators, introducing a potential source of inter-observer variability. Despite adherence to a standardised protocol, US is inherently operator-dependent and may be subject to variability in both interpretation and technique. However, in an interim analysis, we observed excellent reliability for both inter-rater and intra-rater comparisons, with Intraclass Correlation Coefficients of 0.91 and 0.94, respectively. On the other hand, several biomarkers were missed in the follow-up, limiting the potency of the study to demonstrate significant changes. The lack of a control group is another limitation of our study; however, preliminary reports suggest that cancer and its treatments may contribute to the observed changes in body composition [70].

Despite the limitations mentioned above, this study has several strengths. Firstly, to our knowledge, it is the first study to use US as a tool to analyse body composition in women with breast cancer, representing an innovative and non-invasive approach. Moreover, the prospective cohort design will enable robust future analyses of the evolution of the parameters under investigation.

Given the accessibility and non-invasive nature of tools such as BIA and ultrasound, future studies should aim to validate these techniques for the early detection of pathological body composition phenotypes, including sarcopenic obesity. Longitudinal research with extended follow-up periods would be valuable to determine whether these early alterations persist, progress, or respond to targeted interventions. Further investigations should also explore the predictive value of early changes in body composition for clinical outcomes such as cardiovascular events, frailty, and reduced quality of life; the impact of menopausal status, tumor subtype, and specific cancer treatments (e.g., taxanes, corticosteroids, endocrine therapy) on body composition trajectories; the effectiveness of early lifestyle interventions—particularly individualized nutrition and physical activity programs—in reversing or mitigating these adverse changes; and the integration of body composition metrics into risk stratification models to inform survivorship care planning. A deeper understanding of these dynamics could inform proactive strategies to preserve muscle function, reduce fat-related metabolic risk, and ultimately improve long-term outcomes in breast cancer survivors. AI-based software capable of analyzing muscle echogenicity will enable us to evaluate changes in muscle quality more accurately and will strengthen the validity of our complementary measurements.

This preliminary study has helped guide the design of future analyses and the selection of variables currently being collected. We share these findings with the scientific community because we believe that the combined use of BIA and ultrasound represents a feasible and rigorous approach to studying body composition in breast cancer patients, particularly in clinical settings where CT scans are not routinely performed at diagnosis, unlike in some other cancer types. We hope this encourages other research groups to explore this area in greater depth, employing methodologies aligned with current guidelines for body composition assessment and emerging phenotypes, such as sarcopenic obesity.

## 5. Conclusions

In women with recently diagnosed breast cancer, an increase in body fatness and a decrease in some muscle mass and function-derived parameters were observed after one year of follow-up. Although no significant changes were found in metabolic biomarkers related to glucose and lipid metabolism, inflammation, and muscle biology, their relationship with body composition changes warrants further investigation, as they may be involved in the early onset of metabolic disturbances. As the evolution of BMI and body weight showed significant weak correlations with changes in lipid metabolism, vitamin D, and myostatin, the changes in FM and in its distribution were associated with glucose metabolism and with the evolution of HDL-cholesterol. Therefore, the assessment of body composition and function in women with breast cancer may contribute to the development of prognosis algorithms dealing with metabolic and functional evolution that may be reversed with targeted therapy.

## Figures and Tables

**Figure 1 nutrients-17-02487-f001:**
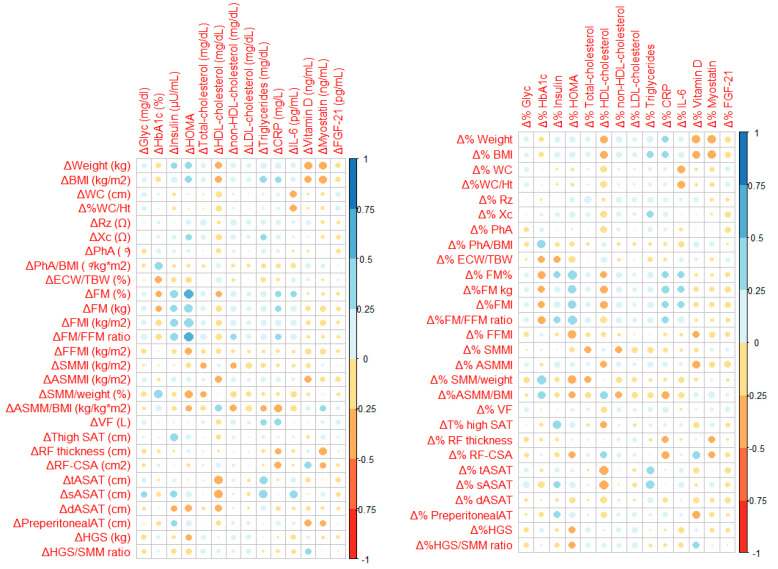
Correlations between biomarkers and body composition parameters. Absolute differences are shown on the left; relative differences on the right. The plot highlights positive correlations in blue, negative correlations in red, and weaker correlations in lighter shades. Large circles indicate a strong correlation. Small circles indicate a weak or negligible correlation. Spearman’s rho correlation coefficients and their *p*-values are described in Appendix A. ASMM: appendicular skeletal muscle mass; ASMMI: appendicular skeletal muscle mass index; AT: adipose tissue; BIA: bioelectrical impedance analysis; BMI: body mass index; CRP: C-reactive protein; dASAT: deep abdominal sub-cutaneous adipose tissue; ECW: extracellular water; FGF-21: fibroblast growth factor 21; FFM: fat-free mass; FFMI: fat-free mass index; FM: fat mass; FMI: fat mass index; Glyc: glycemia; HbA1c: glycosylated haemoglobin; HDL: high-density lipoprotein; HGS: handgrip strength; HOMA-IR: insulin resistance homeostatic model assessment; IL-6: interleukin 6; LDL: low density lipoproteins; PhA: phase angle; RF: rectus femoris; RF-CSA: rectus femoris cross-sectional area; Rz: resistance; SAT: subcutaneous adipose tissue; sASAT: superficial abdominal subcutaneous adipose tissue; SMM: skeletal muscle mass; SMMI: skeletal muscle mass index; tASAT: total abdominal subcutaneous adipose tissue; TBW: total body water; US: ultrasound; VF: visceral fat; WC: waist circumference; WHtR: waist-to-height ratio; Xc: reactance.

**Table 1 nutrients-17-02487-t001:** Baseline characteristics of the sample.

	Total (n = 61)	Premenopausal Status (n = 16)	Menopausal Status (n = 45)	*p*-Value
Age (years)	58.3 (10.0)	47.2 (3.0)	62.3 (8.6)	<0.001
Tumor subtype				
In situ	13 (21.3)	3 (18.8)	10 (22.2)	
Luminal A and B	41 (67.2)	12 (75.0)	29 (64.4)	
Her 2+ (Luminal B HER2+ and HER2+)	4 (6.6)	1 (6.3)	3 (6.6)	0.962
Triple negative	1 (1.6)	0 (0.0)	1 (2.2)	
Unknown	2 (3.3)	0 (0.0)	2 (4.4)	
Treatment (after surgery)				
ChT	14 (23.0)	4 (25.0)	10 (22.2)	1.000
RT	45 (73.8)	14 (87.5)	31 (68.9)	0.196
HT	52 (85.3)	14 (87.5)	38 (84.4)	1.000
Tamoxifen	22 (36.1)	12 (75.0)	10 (22.2)	0.014
Letrozole	22 (36.1)	2 (12.5)	20 (44.4)	
Anastrozole	8 (13.1)	0 (0.0)	8 (17.8)	
Smoker				
Never	53 (86.9)	15 (93.8)	38 (84.4)	
Currently smoke	5 (8.2)	1 (6.3)	4 (8.9)	0.526
Ex-smoker	3 (4.9)	0 (0.0)	3 (6.7)	
Hypertension/dyslipidaemia	23 (37.7)	4 (25.0)	19 (42.2)	0.222
Diabetes mellitus	3 (4.9)	1 (6.3)	2 (4.4)	0.082

Data are presented as n (percentages); age is expressed as mean (standard deviation). ChT: chemotherapy; HER2: human epidermal growth factor receptor 2; HT: hormone therapy; RT: radiation therapy.

**Table 2 nutrients-17-02487-t002:** Evolution of anthropometric measurements, strength, and body composition parameters by bioimpedance (n = 61).

	Basal	One Year	Absolute Change	Relative Change (%)	*p*-Value
Weight (kg)	62.6 (12.3)	63.9 (11.7)	1.0 (−0.7; 3.9)	1.4 (−1.2; 6.0)	0.021
BMI (kg/m^2^)	23.9 (4.6)	25.3 (4.4)	0.4 (−0.3; 1.5)	1.4 (−1.3; 6.0)	0.014
WC (cm)	87.7 (12.2)	89.7 (13.6)	0.6 (−3.0; 8.0)	0.7 (−3.3; 8.4)	0.083
WHtR (cm/cm)	0.55 (0.08)	0.56 (0.09)	0.00 (−0.02; 0.05)	0.73 (−3.30; 8.43)	0.090
Rz (ohm)	692.56 (81.02)	686.36 (81.22)	−9.48 (−32.16; 21.54)	−1.27 (−4.52; 3.40)	0.271
Xc (ohm)	56.13 (7.89)	53.88 (7.89)	−2.21 (−5.24; 0.85)	−3.83 (−8.39; 1.66)	0.001
PhA (^o^)	4.70 (4.15; 5.05)	4.60 (4.25; 4.90)	−0.10 (−0.30; 0.05)	−2.08 (−6.00; 0.98)	0.002
PhA/BMI (^o^/kg∗m^2^)	0.19 (0.04)	0.18 (0.03)	−0.01 (0.02)	−4.90 (8.07)	<0.001
ECW/TBW (%)	46.23 (45.27; 47.90)	47.19 (45.89; 48.50)	0.65 (0.03; 1.48)	1.36 (0.06; 3.21)	<0.001
FM (kg)	25.02 (8.69)	26.22 (8.36)	1.03 (−0.85; 3.56)	3.58 (−2.68; 15.22)	0.017
FM (%)	38.92 (6.99)	40.14 (6.64)	0.90 (−1.16; 3.02)	2.20 (−2.39; 7.72)	0.015
FMI (kg/m^2^)	9.53 (6.68; 12.86)	9.99 (7.82; 12.42)	0.43 (−0.28; 1.41)	3.62 (−2.54; 15.34)	0.006
FM/FFM ratio	0.63 (0.50; 0.81)	0.67 (0.55; 0.84)	0.02 (−0.03; 0.09)	4.37 (−3.52; 12.75)	0.018
FFMI (kg/m^2^)	14.99 (2.08)	14.96 (1.63)	0.18 (−0.27; 0.40)	1.16 (−0.27; 2.99)	0.848
SMMI (kg/m^2^)	6.22 (0.72)	6.13 (0.76)	−0.08 (−0.33; 011)	−1.15 (−5.16; 1.83)	0.035
ASMMI (kg/m^2^)	5.44 (5.09; 6.02)	5.48 (5.16; 5.78)	0.04 (−0.11; 017)	0.75 (−2.06; 3.18)	0.215
ASMM/BMI (kg/kg∗m^2^)	0.57 (0.07)	0.56 (0.07)	−0.01 (−0.02; 0.00)	−1.64 (−3.92; 0.60)	0.002
SMM/weight (%)	26.45 (2.83)	25.69 (3.04)	−0.75 (−1.54; −0.06)	−2.90 (−5.70; −0.24)	<0.001
VF (L)	1.55 (1.20; 2.42)	1.67 (1.17; 2.41)	0.05 (−0.29; 0.41)	2.95 (−16.69; 22.02)	0.549
HGS (kg)	23.4 (4.5)	22.3 (5.7)	−1.0 (3.0)	−3.9 (12.5)	0.110
HGS/SMM (kg/kg)	1.45 (4.5)	1.39 (0.28)	−0.06 (0.21)	−3.02 (14.66)	0.037

Data are expressed as means (standard deviation) or medians (interquartile range). ASMMI: appendicular skeletal muscle mass index; BMI: body mass index; ECW: extracellular water; FFMI: fat-free mass index; FM: fat mass; HGS: handgrip strength; PhA: phase angle; Rz: resistance; SMM: skeletal muscle mass; TBW: total body water; VF: visceral fat; WC: waist circumference; WHtR: waist–to–height ratio; Xc: reactance.

**Table 3 nutrients-17-02487-t003:** Evolution of body composition parameters by ultrasound (n = 53).

	Basal (n = 58)	One Year (n = 55)	Absolute Change	Relative Change (%)	*p*-Value
Abdominal measurements					
tASAT (cm)	1.61 (0.67)	1.80 (0.67)	0.10 (−0.08; 0.44)	7.03 (−5.20; 27.10)	0.011
sASAT (cm)	0.71 (0.30)	0.80 (0.33)	0.01 (−0.09; 0.19)	1.24 (−15.42; 35.00)	0.097
dASAT (cm)	0.90 (0.44)	1.02 (0.43)	0.15 (−0.06; 0.31)	14.44 (−7.43; 48.65)	0.020
Preperitoneal AT (cm)	0.59 (0.36; 0.82)	0.70 (0.51; 0.97)	0.09 (−0.09; 0.29)	15.19 (−14.12; 63.38)	0.031
Thigh and RF measurements					
Thigh SAT (cm)	1.04 (0.36)	1.29 (0.43)	0.25 (0.11; 0.37)	24.16 (0.14; 39.15)	<0.001
Thickness RF (cm)	1.04 (0.91; 1.26)	1.18 (1.01; 1.40)	0.12 (0.01; 0.29)	11.05 (1.10; 30.62)	<0.001
RF-CSA (cm^2^)	3.10 (0.81)	3.45 (0.96)	0.36 (0.76)	14.09 (25.25)	0.001

Data are expressed as means (standard deviation) or medians (interquartile range). AT: adipose tissue; dASAT: deep abdominal subcutaneous adipose tissue; RF: rectus femoris; RF-CSA: rectus femoris cross-sectional area; sASAT: superficial abdominal subcutaneous adipose tissue; SAT: subcutaneous adipose tissue; tASAT: total abdominal subcutaneous adipose tissue.

**Table 4 nutrients-17-02487-t004:** Evolution of biomarkers.

	Basal	One Year	Absolute Change	Relative Change (%)	*p*-Value
Glycemia (mg/dL)	96.0 (89.0; 102.0)	96.5 (91.0; 103.8)	1.5 (−7.8; 8.0)	1.7 (−7.3; 8.9)	0.548 (n = 60)
HbA1c (%)	5.6 (5.4; 5.8)	5.6 (5.4; 5.8)	0.0 (−0.1; 0.2)	0.0 (−1.7; 3.4)	0.076 (n = 57)
Insulin (µUI/mL)	7.4 (4.1; 15.8)	7.9 (3.6; 15.6)	0.2 (−4.7; 5.4)	0.6 (−43.3; 80.3)	0.797 (n = 49)
HOMA-IR	1.8 (1.2; 3.9)	2.4 (1.2; 4.9)	2.4 (1.2; 4.9)	6.6 (−38.0; 99.7)	0.436 (n = 37)
Total cholesterol (mg/dL)	208.8 (38.1)	200.6 (40.6)	−7.6 (35.1)	−2.6 (±17.5)	0.109 (n = 56)
HDL-cholesterol (mg/dL)	67.4 (14.0)	66.1 (33.9)	−1.0 (7.7)	−0.5 (±11.2)	0.334 (n = 55)
Non-HDL-cholesterol (mg/dL)	140.5 (34.0)	133.5 (35.6)	−9.0 (−27.0; 12.0)	−7.0 (−18.7; 7.9)	0.159 (n = 55)
LDL-cholesterol (mg/dL)	118.9 (36.4)	112 (38.5)	−9.0 (−25.0; 10.0)	−9.0 (−25.0; 10.0)	0.116 (n = 55)
Triglycerides (mg/dL)	92.0 (72.0; 129.5)	92.0 (74.0; 136.0)	7.0 (−26.0; 31.8)	7.9 (−28.4; 31.4)	0.701 (n = 56)
CRP (mg/L)	2.80 (2.80; 3.20)	2.80 (2.85; 3.00)	0.00 (−0.10; 0.00)	0.00 (−3.45; 0.00)	0.172 (n = 51)
IL-6 (pg/mL)	1.80 (1.40; 3.50)	1.40 (1.40; 2.70)	0.00 (−1.10; 0.50)	0.00 (−44.00; 21.74)	0.397 (n = 51)
Vitamin D (ng/mL)	24.0 (14.0; 33.5)	25.0 (17.0; 38.0)	1.0 (−4.5; 7.5)	3.8 (−23.1; 46.2)	0.226 (n = 53)
Myostatin (ng/mL)	33.03 (27.95; 39.02)	33.07 (29.67; 37.84)	0.60 (−3.50; 3.54)	1.60 (−8.23; 11.79)	0.968 (n = 41)
FGF-21 (pg/mL)	68.57 (38.75; 144.19)	92.79 (48.22; 179.20)	14.89 (−39.77; 76.93)	32.13 (−32.69; 100.68)	0.226 (n = 40)

Data are expressed as means (standard deviation) or medians (interquartile range). CRP: C-reactive protein; FGF-21: fibroblast growth factor 21; HbA1c: glycosylated haemoglobin; HDL: high-density lipoprotein; HOMA-IR: insulin resistance homeostatic model assessment; IL-6: interleukin 6; LDL: low density lipoprotein.

## Data Availability

The original contributions presented in the study are included in the article/Appendix A; further inquiries can be directed to the corresponding authors.

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
