# Peer review of "Relationship Between Body Composition and Biomarkers in Adult Females with Breast Cancer: 1-Year Follow-Up Prospective Study"

_nutrients, 2025, doi:10.3390/nu17152487_

Round 1

Reviewer 1 Report

Comments and Suggestions for Authors

The topic is relevant however major methodological limitation characterize the study which need to be improved, more other changes are needed to improve the presentation of the manuscript.

Major comments:

1)    Many variables used from the BIA are not reliable due to the limits of instrument, which are simply approximates estimations and never direct measurements, such as fat mass, skeletal muscle mass, appendicular skeletal muscle mass, visceral fat etc. ARE NOT ACCURATE, and considering them from BIA is a huge methodological limitation. Otherwise authors need to use more accurate body composition techniques. The only value that authors can use from BIA is the phase angle; any other thing is a pure speculation.

2)     Authors in their study used simple correlation between body composition parameters and biomarkers. Correlations are not sufficient, as more complex interaction such as regression model analysis are mandatory needed, or at least partial correlation analysis after accounting for confounders.  

3)    The sample size is small, for the number of variables that have included. Authors should compute are reliable power analysis to justify their sample.

4)    It is not clear if the analysis was conducted on completers only? What is the rate of drop out at one-year follow up. An-intent-to-treat analysis is required. 

5)    The discussion section should be structured as follows:

  • The main findings of the study, and their comparison with previously published studies on the topic.
  • The clinical implication of study should be clearly stated based on the findings with out speculation
  • The strengths and limitations should be extensively described
  • The new direction for future research still needed on this topic

Other comments

1)    The title need to be changed to read as: Relationship between Body composition and biomarkers in adult females with breast            cancer: 1-year follow-up prospective study.   

2)    The abstract is extremely long and should be absolutely shortened to meet the journal guidelines

3)    Figure 1 should be remove 

Author Response

Reviewer 1:

Comments and Suggestions for Authors

The topic is relevant however major methodological limitation characterize the study which need to be improved, more other changes are needed to improve the presentation of the manuscript.

Major comments:

1)    Many variables used from the BIA are not reliable due to the limits of instrument, which are simply approximates estimations and never direct measurements, such as fat mass, skeletal muscle mass, appendicular skeletal muscle mass, visceral fat etc. ARE NOT ACCURATE, and considering them from BIA is a huge methodological limitation. Otherwise authors need to use more accurate body composition techniques. The only value that authors can use from BIA is the phase angle; any other thing is a pure speculation.

Thank you very much for your comments. We agree with your observations, as many BIA-derived parameters are calculated using formulas developed in specific healthy populations, which may not apply to ill individuals whose water distribution may differ from the standard. Moreover, these formulas are particularly inaccurate at the extremes of BMI.

However, the most recent ASPEN guidelines¹ state: “BIA is a practical, portable, non-invasive tool that poses minimal risks and low costs relative to the other body composition assessment methods. These characteristics allow its use in any setting, such as epidemiological and clinical studies, and render BIA an ideal assessment technique for follow-up studies, in which repeated measurements are necessary and easily obtained. (…) An acceptable mean level accuracy for BIA assessments has been established in healthy, nonobese participants.”

Therefore, in a clinical study such as this, BIA may provide an opportunity to move beyond the simplistic measurement of BMI. Raw parameters such as phase angle, resistance, and reactance should be described as markers of cellular status and their relationship with hydration. However, in this exploratory study, associations between these parameters and changes in biochemical biomarkers were not generally observed. Some associations were found with changes in body compartments, such as body fat and skeletal muscle mass, when corrected by adiposity parameters such as SMM/weight or ASMM/BMI. This supports biological plausibility and, to a certain extent, validates the use of this method of body composition assessment in the study.

Furthermore, the volunteers enrolled at baseline were women with recently diagnosed breast cancer not yet affected by cancer treatment. Although the study included older adults, only 8.2% (n=5) of the sample were aged 75 years or older. The BMI range at baseline was 15.4 to 38.6 kg/m², and 5 individuals (8.2%) had BMI values below 18.5 or above 34 kg/m². At the end of follow-up, 9.8% (n=6) were over 75 years old, and 6.6% (n=4) were at the extremes of BMI. It is possible that BIA parameters were less accurate in these subgroups. Additionally, overestimation of FM and SMM may have occurred after one year of therapy, considering that increases in body water are commonly seen with disease progression, as noted in the discussion section: “This increase in TBW represents a limitation of BIA, as it may lead to an underestimation of FM and an overestimation of FFM in these patients.” In any case, our study did not show changes in crude data related to FFM and ASMMI, whereas changes were observed in FM-related metrics, thus allowing BIA analysis to differentiate between body compartments not described by anthropometric measurements.

Of course, CT or MRI could provide more precise tools for body composition assessment. However, CT is not always available in breast cancer follow-up, and MRI is not commonly used in the clinical management of this disease. We are currently studying CT-based muscle and fat changes in those cases where this approach is possible, and we hope to present reliable data in the future.

We have added the following statement to the limitations section: “On the other hand, BIA analysis may not be accurate enough to estimate body compartments in patients at the extremes of age or BMI, or with altered hydration status. Nevertheless, the changes observed during follow-up were consistent with previously published studies [44], and although an increase in ECW/TBW was observed, the percentage and absolute values of TBW did not change in a clinically relevant manner (from 28.33 ± 3.82 L to 28.40 ± 3.63 L, and from 45.69 ± 4.92% to 44.75 ± 4.74%). More studies are necessary to assess the external validity of BIA-derived data in breast cancer.”

1.- Sheean P, Gonzalez MC, Prado CM, McKeever L, Hall AM, Braunschweig CA. American Society for Parenteral and Enteral Nutrition Clinical Guidelines: The Validity of Body Composition Assessment in Clinical Populations. JPEN J Parenter Enteral Nutr. 2020 Jan;44(1):12-43. doi: 10.1002/jpen.1669. Epub 2019 Jun 19. PMID: 31216070.

2)     Authors in their study used simple correlation between body composition parameters and biomarkers. Correlations are not sufficient, as more complex interaction such as regression model analysis are mandatory needed, or at least partial correlation analysis after accounting for confounders

Thank you again for your valuable comments. Several variables may act as confounders or effect modifiers, but the sample size is not large enough to allow for proper adjustment. We have now explored the significant associations identified in the correlation analysis to determine possible adjustments, using rank transformation of the data since not all variables satisfied the assumption of normality. Among the variables considered, only age was not affected by collinearity. The crude and age-adjusted models are presented in Supplementary Table 3.

Supplementary table 3: Multiple linear regression models used to explain the relationship between changes in body composition parameters (independent variable) and the evolution of biomarkers (dependent variable): crude models and adjusted for age.

Associated rank-transformed variables

Standardized β; p

DWeight - DHDL-cholesterol

Crude:              -0.319; 0.018

Adjusted:         -0.298; 0.030

DWeight - DVitamin D

Crude:              -0.339; 0.013

Adjusted:         -0.356; 0.011

DWeight - DMyostatin

Crude:              -0.360; 0.021

Adjusted:         -0.348; 0.027

DBMI - DHDL-cholesterol

Crude:              -0.289; 0.034

Adjusted:         -0.296; 0.030

DBMI - DTriglycerides

Crude:              0.272; 0.043

Adjusted:         0.302; 0.028

DBMI - DVitamin D

Crude:              -0.308; 0.025

Adjusted:         -0.322; 0.022

DBMI - DMyostatin

Crude:              -0.360; 0.021

Adjusted:         -0.348; 0.027

DWC  -DIL-6

Crude:              -0.322; 0.021

Adjusted:         -0.318; 0.024

DWC/Ht-DIL-6

Crude:              -0.317; 0.024

Adjusted:         -0.313; 0.027

DPhA/BMI  - DHbA1c

Crude:              0.386; 0.007

Adjusted:         0.415; 0.004

DECW/TBW -DHbA1c

Crude:              -0.311; 0.033

Adjusted:         -0.314; 0.032

DFM  -DInsulin

Crude:              0.330; 0.020

Adjusted:         0.303; 0.039

DFM - DHOMA-IR

Crude:              0.503; 0.002

Adjusted:         0.466; 0.006

DFM - DInsulin

Crude:              0.327; 0.022

Adjusted:         0.302; 0.037

DFM  - DHOMA-IR

Crude:              0.436; 0.017

Adjusted:         0.390; 0.020

DFMI  -DInsulin

Crude:              0.341; 0.017

Adjusted:         0.316; 0.030

DFMI - DHOMA-IR

Crude:              0.461; 0.004

Adjusted:         0.417; 0.013

DVF  - DCRP

Crude:              0.305; 0.035

Adjusted:         0.308; 0.035

DThigh SAT  -DInsulin

Crude:              0.349; 0.025

Adjusted:         0.324; 0.035

DtASAT  - DHDL-cholesterol

Crude:              -0.447; 0.002

Adjusted:         -0.444; 0.002

DtASAT - DTriglycerides

Crude:              0.381; 0.008

Adjusted:         0.382; 0.008

DsASAT  - DHDL-cholesterol

Crude:              -0.398; 0.006

Adjusted:         -0.395; 0.005

DsASAT - DTriglycerides

Crude:              0.405; 0.004

Adjusted:         0.405; 0.005

DsASAT  - DIL-6

Crude:              0.340; 0.024

Adjusted:         0.340; 0.026

DPreperitonealAT -DVitamin D

Crude:              -0.298; 0.046

Adjusted:         -0.299; 0.049

DFFMI  -DHOMA-IR

Crude:              -0.333; 0.044

Adjusted:         -0.328; 0.042

DSMMI - DTotal-cholesterol

Crude:              -0.261; 0.052

Adjusted:         -0.255; 0.053

DSMMI - Dnon-HDL-cholesterol

Crude:              -0.260; 0.055

Adjusted:         -0.260; 0.053

DASMMI  -DHDL-cholesterol

Crude:              -0.279; 0.039

Adjusted:         -0.263; 0.053

DASMMI  -DVitamin D

Crude:              -0.303; 0.027

Adjusted:         -0.309; 0.026

 D RF thickness - DMyostatin

Crude:              -0.288; 0.084

Adjusted:         -0.288; 0.087

DFM/FFM ratio -DInsulin

Crude:              0.358; 0.012

Adjusted:         0.335; 0.020

DFM/FFM ratio -DHOMA-IR

Crude:              0.524; 0.001

Adjusted:         0.487; 0.003

DSMM/weight - DHbA1c

Crude:              0.414; 0.004

Adjusted:         0.477; 0.001

DSMM/weight  - DHOMA-IR

Crude:              -0.335; 0.042

Adjusted:         -0.286; 0.089

DSMM/weight -DTotal-cholesterol

Crude:              -0.268; 0.046

Adjusted:         -0.326; 0.014

DASMM/BMI  -DHDL-cholesterol

Crude:              0.274; 0.043

Adjusted:         0.253; 0.065

DASMM/BMI  -Dnon-HDL-cholesterol

Crude:              -0.275; 0.042

Adjusted:         -0.316; 0.019

DASMM/BMI  -DCRP

Crude:              -0.352; 0.011

Adjusted:         -0.344; 0.015

D%Weight -D%HDL-cholesterol

Crude:              -0.338; 0.012

Adjusted:         -0.319; 0.019

D%Weight - D%Vitamin D

Crude:              -0.373; 0.006

Adjusted:         -0.389; 0.005

D%Weight - D%Myostatin

Crude:              -0.335; 0.033

Adjusted:         -0.321; 0.041

D%BMI - D%HDL-cholesterol

Crude:              -0.330; 0.014

Adjusted:         -0.310; 0.022

D%BMI - D% Triglycerides

Crude:              0.268; 0.046

Adjusted:        0.290; 0.034

D%BMI - D%Vitamin D

Crude:              -0.330; 0.016

Adjusted:         -0.344; 0.014

D%BMI - D%Myostatin

Crude:              -0.355; 0.023

Adjusted:         -0.342; 0.029

D%WC - D%IL-6

Crude:              -0.326; 0.019

Adjusted:        -0.325; 0.022

D%WC/Ht -  D%IL-6

Crude:              -0.326; 0.019

Adjusted:        -0.325; 0.022

D%Xc - D% Triglycerides

Crude:              0.284; 0.034

Adjusted:        0.284; 0.035

D%PhA/BMI - D%HbA1c

Crude:              0.397; 0.006

Adjusted:        0.427; 0.003

D%FM_% - D%HbA1c

Crude:             -0.324; 0.026

Adjusted:       -0.450; 0.003

D%FM_% - D%Insulin

Crude:              0.278; 0.053

Adjusted:        0.241; 0.104

D%FM_% -D%HOMA

Crude:              0.452; 0.005

Adjusted:        0.400; 0.020

D%FM_% -D%HDL-cholesterol

Crude:              -0.282; 0.037

Adjusted:         -0.256; 0.069

D%FM_kg - D%HbA1c

Crude:             -0.295; 0.044

Adjusted:       -0.391; 0.010

D%FM_kg -D%HOMA

Crude:              0.362; 0.028

Adjusted:        0.301; 0.077

D%FM_kg -D%HDL-cholesterol

Crude:              -0.288; 0.033

Adjusted:         -0.264; 0.057

D%FM_kg - D%CRP

Crude:              0.306; 0.029

Adjusted:        0.297; 0.039

D%FMI  -D%HOMA

Crude:              0.430; 0.008

Adjusted:        0.375; 0.026

D%FMI -D%HDL-cholesterol

Crude:              -0.273; 0.044

Adjusted:         -0.246; 0.078

D%FMI - D%CRP

Crude:              0.237; 0.094

Adjusted:        0.226; 0.120

D%Thigh SAT -D%Insulin

Crude:              0.362; 0.020

Adjusted:        0.346; 0.022

D%tASAT - D%HDL-cholesterol

Crude:              -0.460; 0.001

Adjusted:         -0.456; 0.001

D%tASAT - D% Triglycerides

Crude:              0.329; 0.023

Adjusted:         0.329; 0.024

D%sASAT - D%HDL-cholesterol

Crude:              -0.410; 0.004

Adjusted:         -0.408; 0.004

D%sASAT - D% Triglycerides

Crude:              0.368; 0.010

Adjusted:         0.368; 0.011

D%PreperitonealAT - D%Vitamin D

Crude:              -0.293; 0.051

Adjusted:         -0.297; 0.052

D%FFMI - D%HOMA

Crude:              -0.351; 0.033

Adjusted:        -0.343; 0.032

D%ASMMI -D%HDL-cholesterol

Crude:              -0.278; 0.040

Adjusted:         -0.263; 0.053

D%ASMMI -D%Vitamin D

Crude:              -0.339; 0.013

Adjusted:         -0.345; 0.013

D%RF-CSA - D%CRP

Crude:              -0.325; 0.036

Adjusted:         -0.316; 0.045

D%RF-CSA -D%Vitamin D

Crude:              0.309; 0.041

Adjusted:         0.317; 0.038

D%SMM/weight  - D%HbA1c

Crude:              0.460; 0.001

Adjusted:         0.513; <0.001

D%SMM/weight - D%HOMA

Crude:              -0.372; 0.023

Adjusted:         -0.327; 0.046

D%ASMM/BMI -D%HbA1c

Crude:              0.315; 0.031

Adjusted:         0.379; 0.011

D%ASMM/BMI -D%HDL-cholesterol

Crude:              0.284; 0.035

Adjusted:         0.265; 0.052

D%ASMM/BMI -D%non-HDL-cholesterol

Crude:              -0.264; 0.051

Adjusted:         -0.298; 0.028

D%ASMM/BMI -D%CRP

Crude:              -0.340; 0.015

Adjusted:         -0.334; 0.018

The variables were rank transformed to be included in the models. ASMM: Appendicular Skeletal Muscle Mass; ASMMI: Appendicular Skeletal Muscle Mass Index; AT: Adipose Tissue; BMI: Body Mass Index; CRP: C-reactive protein; ECW: Extracellular Water; FFM: Fat-Free Mass; FFMI: Fat-Free Mass Index; FM: Fat Mass; FMI: Fat Mass Index; HbA1c: glycosylated haemoglobin; HDL: High-density lipoprotein; HOMA-IR: Insulin Resistance Homeostatic Model Assessment; IL-6: Interleukin 6; PhA: Phase Angle; RF: rectus femoris; RF-CSA: rectus femoris cross-sectional area; SAT: subcutaneous adipose tissue; sASAT: superficial abdominal subcutaneous adipose tissue; SMM: Skeletal Muscle Mass; SMMI: Skeletal Muscle Mass  index; tASAT: total abdominal subcutaneous adipose tissue; TBW: Total Body Water; VF: Visceral Fat; WC: Waist Circumference; WHtR: Waist–to–Height Ratio; Xc: Reactance.

As the Spearman rho coefficient helped us exploratorily understand the associations between changes in body composition parameters and biochemical biomarkers, this new analysis allowed us to examine the average change in ranks of the dependent variables (biochemical biomarkers) for each one-rank increase in the studied body composition parameters (independent variables), adjusting for age.

When examining absolute differences, almost all relationships remained statistically significant and retained the same direction of association, as reflected by the standardized beta coefficients after adjustment. However, statistical significance was lost after adjustment in the following pairs: DASMMI (kg/m2) -DHDL-cholesterol (mg/dL), DSMM/weight (%) - DHOMA-IR, and DASMM/BMI (kg/kg*m2) -DHDL-cholesterol (mg/dL).

In the univariate analysis, associations between  DSMMI (kg/m2) - DTotal-cholesterol (mg/dL), DSMMI (kg/m2) - Dnon-HDL-cholesterol (mg/dL), and D RF thickness (cm) - DMyostatin (ng/mL) were not observed, suggesting that the correlations previously detected may represent more spurious associations.

When relative (%) differences where explored, age acted as a confounder in the following pairs: D%FM_% -D%HDL-cholesterol, D%FM_kg -D%HOMA, D%FM_kg -D%HDL-cholesterol, D%FMI -D%HDL-cholesterol, D%ASMMI -D%HDL-cholesterol, D%ASMM/BMI (kg/kg*m2) -D%non-HDL-cholesterol, and D%ASMM/BMI (kg/kg*m2) -D%HDL-cholesterol.

No significant associations were found in the univariate analysis for these pairs: D%FM_% - D%Insulin, D%FMI - D%CRP, and D%PreperitonealAT - D%Vitamin D.

We have added these paragraphs to the manuscript:

  1. Statistical section: “To account for confounders in the associations between changes in body composition parameters and biochemical biomarkers, we performed both crude and age-adjusted linear regression analyses using rank-transformed variables that showed statistically significant associations in the correlation analysis. Since the body composition parameters (independent variables) were interrelated and given the limited sample size, only age (rank-transformed) was included as a covariate.”
  2. Results section:

b.1. Title: “Association” (instead of Correlations) between body composition changes and biomarkers evolution

b.2. “The crude and age-adjusted linear regression models are presented in the supplementary table 3. Robust and biologically plausible associations were observed between body composition parameters and key metabolic biomarkers after adjustment for age. Increased FM, as measured by DFM (%), D%FM_% DFMI, DFM/FFM ratio, showed strong positive associations with DHOMA-IR and D%HOMA (adjusted β ranging from 0.400 to 0.487; p<0.025). Increased SAT, as measured by DtASAT, D%tASAT and D%sASAT demonstrated strong negative associations with DHDL-cholesterol and D%HDL-cholesterol (adjusted β ranging from 0.408 to 0.456; p<0.005). In addition, DsASAT was significantly and positively associated with DTriglycerides (adjusted β  0.405; p=0.005). However, some of the studied associations were no longer significant after adjusting for age.” 

  1. Discussion section:

c.1. “Our analyses revealed several robust and statistically significant associations between changes in body composition, classical anthropometric parameters and biochemical biomarkers, independent of age. Notably, increases in FM—expressed as percentage FM and FMI—were strongly and positively associated with markers of insulin resistance, including HOMA-IR. Conversely, SAT in thigh and abdominal regions showed consistent inverse relationships with HDL-cholesterol and positive associations with triglycerides, highlighting their impact on lipid metabolism.“

c.2. “The results of both crude and age-adjusted linear regression models revealed several statistically significant associations between changes in body composition, classical anthropometric parameters, and biochemical biomarkers. However, some of these associations were no longer significant after adjusting for age. This attenuation of significance after adjustment suggests that age influences these associations and may partly explain the correlations observed in the crude analyses.”

3)    The sample size is small, for the number of variables that have included. Authors should compute are reliable power analysis to justify their sample.

In fact, this is an important consideration that depends on the primary objective of the study. In an exploratory study such as this, the ability to detect differences in the parameters assessed after one year of follow-up may vary according to the specific parameter under analysis. In the larger project of which this study forms a part, the primary objective is to describe changes in adiposity, and its distribution over time, following breast cancer diagnosis, as well as to explore factors influencing these changes. For example, to specifically detect a change in percent fat mass (%FM), a sample size of 34 or 46 paired observations would be required to achieve 80% or 90% statistical power, respectively, with a two-sided significance level of 5%, assuming a mean paired difference of 2% and a standard deviation of 4 %.

In Supplementary Table 4, we present the achieved power for comparisons that were not statistically significant, including anthropometric, body composition, strength, and biochemical biomarker variables.

Supplementary table 4: Post-hoc power analysis for not-significant changes after one year of follow-up

n

Cohen’s d for paired samples

Power obtained

WC (cm)

61

0.23

0.424

WHtR (cm/cm)

61

0.22

0.394

Rz (Ohm)

61

0.14

0.190

FFMI (kg/m2)

61

0.03

0.056

ASMMI (kg/m2)

61

0.11

0.135

VF (L)

58

0.06

0.073

HGS (kg)

61

0.31

0.664

sASAT (cm)

53

0.23

0.376

Glycemia (mg/dL)

60

0.02

0.053

HbA1c (%)

47

0.23

0.339

Insulin (µUI/mL)

49

0.14

0.161

HOMA-IR

37

0.10

0.091

Total cholesterol (mg/dL)

56

0.22

0.366

HDL-cholesterol (mg/dL)

55

0.13

0.157

Non-HDL-cholesterol (mg/dL)

55

0.19

0.283

LDL-cholesterol (mg/dL)

55

0.22

0.361

Triglycerides (mg/dL)

56

0.05

0.066

CRP (mg/L)

51

0.21

0.313

IL-6 (pg/mL)

51

0.06

0.070

Vitamin D (ng/mL)

53

0.23

0.376

Myostatin (ng/mL)

41

~0.00

0.050

FGF-21 (pg/mL)

40

0.22

0,274

ASMMI: Appendicular Skeletal Muscle Mass Index; CRP: C-reactive protein; FGF-21: Fibroblast growth factor 21; FFMI: Fat-Free Mass Index; Glyc: Glycemia; HbA1c: glycosylated haemoglobin; HDL: High-density lipoprotein; HGS: Handgrip strength; HOMA-IR: Insulin Resistance Homeostatic Model Assessment; IL-6: Interleukin 6; LDL: Low density lipoproteins; Rz: Resistance; sASAT: superficial abdominal subcutaneous adipose tissue; VF: Visceral Fat; WC: Waist Cir-cumference; WHtR: Waist–to–Height Ratio

Cohen's d: difference between two means divided by the pooled standard deviation (effect size: 0.2 = small effect; 0.5 = medium effect; 0.8 = large effect). Power was calculated with jPower Module from jamovi Cloud (https://cloud.jamovi.org/), assuming a two-sided criterion for detection that allows for a maximum Type I error rate of α = 0.05.

This table presents the results of post hoc power analyses conducted for variables in which changes after one year of follow-up were not statistically significant. For each variable, the observed effect size, sample size, and corresponding statistical power are listed to provide context regarding the study's ability to detect meaningful differences over time. These analyses help clarify that the non-significant findings may be due to insufficient statistical power.

Regarding Spearman correlation analysis for absolute changes, most correlation coefficients were far below 0.500, except for ΔFM (%) vs ΔHOMA-IR (ρ = 0.522, p = 0.001), ΔFMI vs ΔHOMA-IR (ρ = 0.481, p = 0.003), and ΔFM/FFM ratio vs ΔHOMA-IR (ρ = 0.538, p = 0.001). For relative changes (%), the highest levels of rho coefficient were around 0.5 such as Δ%FM/FFM ratio vs Δ%HOMA -ρ = 0.480, p=0.003. Spearman’s rho coefficients below 0.4 generally indicate a low effect size, which reduces statistical power to detect associations given our limited sample size. Consequently, we were able to detect statistically significant associations in variables with larger effect sizes but not in weaker ones. Correlations with rho values greater than 0.35 and sample size around 50 attained statistical power in the range of 0.75 to 0.80, which is considered reasonable for detecting effects. In contrast, correlations with rho values between 0.15 and 0.25 and sample sizes under 50 had power below 0.50, indicating the analyses were underpowered. The strongest correlations (rho >0.5) with sample sizes exceeding 40 achieved power levels above 0.85, indicating robust findings (https://cloud.jamovi.org/a727f1b2-2974-42c8-a435-bc7269973fa4/; data not shown).

The following text has been included in the manuscript:

-Methods,

“As the main project focuses on the evolution of adiposity over time following breast cancer diagnosis, the sample size calculation for this exploratory analysis was based on detecting changes in the most clinically accessible adiposity marker, %FM estimated by BIA. Accordingly, a sample size of 44 paired observations would be required to reliably detect an effect size of δ ≥ 0.5 with a statistical power greater than 0.9, assuming a two-sided test and a maximum Type I error rate of α = 0.05. For US comparisons, the sample size estimate was more conservative due to the lack of established data on minimally clinically relevant changes. In this context, 52 paired observations would be needed to detect an effect size of δ ≥ 0.4 with a power greater than 0.8, also assuming a two-sided test and α = 0.05.”

-Limitations,

“The supplementary table 4 presents the results of post hoc power analyses for variables (body composition, strength and biochemical biomarkers) in which changes after one year of follow-up were not statistically significant. These analyses indicate that the non-significant findings may be attributable to insufficient statistical power, which ranged from 0.050 to 0.664. An increased sample size would be required to detect statistically significant differences. Moderate to large correlations (±0.35 and above) generally achieved acceptable power (>0.70), whereas weaker correlations (<0.30) were underpowered. Therefore, the lack of significance in correlations with low rho values (approximately 0.10–0.20) may also be due to insufficient power. In contrast, significant correlations with moderate to large rho values were adequately powered.

4)    It is not clear if the analysis was conducted on completers only? What is the rate of drop out at one-year follow up. An-intent-to-treat analysis is required. 

Thank you for this comment, as it addresses a very important topic concerning the internal and external validity of the study. At the time of this interim analysis, 138 patients had been enrolled. Of these, 71 had completed one year of follow-up (51.5% of the total sample). The remaining women were either still awaiting their one-year reassessment or had been lost to follow-up. The main challenge to completing the one-year follow-up at that time was the onset of the COVID-19 pandemic, during which all non-essential healthcare activities were suspended. Subsequently, many patients missed follow-up visits due to fear of infection. However, this is an ongoing study with a planned follow-up of five years, and until the initially estimated sample size of 425 women has been reached and their five years of follow-up completed, we cannot provide a definitive description of the dropout rate.

In this analysis, we refined the sample to include only those with all variables required for the one-year follow-up analysis, comprising 44.2% of the initial cohort. Baseline characteristics did not differ significantly between this subset and the full baseline sample. The mean age of the total baseline population was 59.0 ± 10.9 years, with 26.2% being premenopausal. The predominant tumor types were luminal A and B, accounting for 69.3% of cases. Comorbidity rates were similar, with 36.6% of patients presenting hypertension or dyslipidemia. The prevalence of type 2 diabetes differed slightly between the entire baseline sample and the analyzed subset, at 8.4% versus 4.9%, respectively.

We have included this text at the limitations section:

From an initial recruitment of 138 participants, we selected the sample that completed the protocol after one year of follow-up, representing 44.2% of the cohort. Baseline characteristics—including age, hormonal status at diagnosis, tumor type, and comorbidities—were comparable between the total sample and this subset. Although such attrition may affect internal and external validity, the strongest changes and associations observed can be considered in future prospective studies, modeled with additional explanatory variables.

5)  The discussion section should be structured as follows: a) The main findings of the study, and their comparison with previously published studies on the topic; b) The clinical implication of study should be clearly stated based on the findings with out speculation; c) The strengths and limitations should be extensively described; d) The new direction for future research still needed on this topic

Thank you again. Your recommendations have prompted us to reevaluate the content of our paper and improve several sections. Specifically:

  1. We have enhanced the presentation of summarized findings.
  2. We have clarified the implications of the study by making statements less speculative.
  3. We have provided a more thorough discussion of the strengths and limitations.
  4. We have included the following paragraph before the conclusions section:

Given the accessibility and non-invasive nature of tools such as BIA and ultrasound, future studies should aim to validate these techniques for the early detection of pathological body composition phenotypes, including sarcopenic obesity. Longitudinal research with extended follow-up periods would be valuable to determine whether these early alterations persist, progress, or respond to targeted interventions. Further investigations should also explore the predictive value of early changes in body composition for clinical outcomes such as cardiovascular events, frailty, and reduced quality of life; the impact of menopausal status, tumor subtype, and specific cancer treatments (e.g., taxanes, corticosteroids, endocrine therapy) on body composition trajectories; the effectiveness of early lifestyle interventions—particularly individualized nutrition and physical activity programs—in reversing or mitigating these adverse changes; and the integration of body composition metrics into risk stratification models to inform survivorship care planning. A deeper understanding of these dynamics could inform proactive strategies to preserve muscle function, reduce fat-related metabolic risk, and ultimately improve long-term outcomes in breast cancer survivors.

This preliminary study has helped guide the design of future analyses and the selection of variables currently being collected. We share these findings with the scientific community because we believe that the combined use of BIA and ultrasound represents a feasible and rigorous approach to studying body composition in breast cancer patients, particularly in clinical settings where CT scans are not routinely performed at diagnosis, unlike in some other cancer types. We hope this encourages other research groups to explore this area in greater depth, employing methodologies aligned with current guidelines for body composition assessment and emerging phenotypes such as sarcopenic obesity.”

Other comments

1)    The title needs to be changed to read as: Relationship between Body composition and biomarkers in adult females with breast  cancer: 1-year follow-up prospective study.  

Thank you for your suggestion, which has helped to make the manuscript’s message clearer. The title has been changed accordingly.

2)    The abstract is extremely long and should be absolutely shortened to meet the journal guidelines.

Indeed, it was longer than 250 words. Thank you for the advice. We have modified the abstract as follows:

“Background: After diagnosis, it is common for women with breast cancer to gain weight, which is associated with worse clinical outcomes. However, traditional measures such as body weight, BMI, and waist circumference do not detect key changes in body composition, such as fat redistribution or muscle loss. The objective of this exploratory study was to assess the evolution of body composition and muscle strength after one year of treatment, and their relationship with metabolic biomarkers.

Methods: Prospective observational study in newly diagnosed breast cancer patients. Body composition was assessed using bioelectrical impedance analysis (BIA) and ultrasound (US); muscle strength was measured by handgrip dynamometry. Biomarkers analyzed included glucose, insulin, Homeostatic Model Assessment of Insulin Resistance (HOMA-IR), glycosylated hemoglobin (HbA1c), total cholesterol (and its fractions), triglycerides, C-reactive protein (CRP), 6-interleukine (IL-6), vitamin D, myostatin and Fibroblast Growth Factor 21 (FGF-21).

Results: Sixty-one women (mean age 58 years) were included. After one year, fat mass and related parameters significantly increased, while skeletal muscle mass and muscle strength decreased. Sarcopenic obesity prevalence rose from 1.16% to 4.9%. No significant changes were found in biomarkers, but positive correlations were observed between fat parameters and insulin, HOMA-IR, and triglycerides, and negative correlations with HDL-cholesterol.

Conclusions: BIA and US can detect unfavorable changes in body composition that are not reflected in conventional measurements. One-year post-diagnosis, women showed increased fat accumulation, muscle loss, and reduced strength, even without significant metabolic biomarker changes. Further research is warranted to elucidate the long-term clinical implications of these findings.”

3)    Figure 1 should be removed

We understand that the figure may be mistakenly interpreted, leading readers to conclude that some correlations are significant when they are not. This was not the authors' intention. In an exploratory study such as ours, which involves the analysis of multiple variables, the heatmap was intended as a tool for visually identifying relevant associations and summarizing the data in an interpretable manner. The complete data, including all Spearman’s rho correlation coefficients and their corresponding p-values, are provided in Supplementary Tables 1 and 2.

Nevertheless, if the reviewer and editor consider that removing the figure would improve the manuscript, we will accept that decision.

Additionally, we have added the following note as a footnote to the figure:

“Spearman’s rho correlation coefficients and their p-values are described in Supplementary Tables 1 and 2.”

Reviewer 2 Report

Comments and Suggestions for Authors

The article titled " Body composition changes and their relationship with biomarkers evolution one year after breast cancer diagnosis" by Angélica Larrad-Sáinz et al. offers valuable insights. However, several aspects warrant further consideration:

  1. The study’s sample size (N= 61) is relatively small for an exploratory analysis, which may limit statistical power, particularly for subgroup comparisons.
  2. The broad age range (40–80 years) introduces potential confounding, as BC alterations (e.g., sarcopenia, fat redistribution) may differ substantially between younger and older patients. Stratified analyses by age group could clarify these effects.
  3. The absence of a non-cancer control group makes it difficult to discern whether observed BC changes are specific to breast cancer treatment or reflect age-related metabolic shifts
  4. In this study, authors did not measure Leptin/adiponectin or IGF-1, which could provide mechanistic insights into fat redistribution effects.
  5. While BIA is practical in this study, it is less accurate than DEXA or CT/MRI for body composition assessment, particularly in cancer patients with fluid shifts.
  6. While fat parameters correlated with insulin resistance and lipids, the study cannot determine directionality. Does fat gain drive dysregulation, or vice versa?
  7. Authors measured handgrip strength (Jamar dynamometer), which may not fully reflect lower-body or axial muscle function (critical for mobility and metabolic health).
  8. Did patients receive chemotherapy, radiation, or endocrine therapy? For instance, aromatase inhibitors are known to exacerbate muscle loss.
  9. HOMA-IR is derived from fasting glucose/insulin, which may not capture postprandial dysregulation (common in metabolic syndrome).

Author Response

Reviewer 2:

Comments and Suggestions for Authors

The article titled " Body composition changes and their relationship with biomarkers evolution one year after breast cancer diagnosis" by Angélica Larrad-Sáinz et al. offers valuable insights. However, several aspects warrant further consideration:

  1. The study’s sample size (N= 61) is relatively small for an exploratory analysis, which may limit statistical power, particularly for subgroup comparisons.

We agree with this observation. Indeed, the sample size is small to achieve sufficient statistical power to detect certain differences in the studied parameters. We have conducted an extensive post hoc power analysis on the study sample, focusing particularly on the non-significant associations, which we summarize below:

In Supplementary Table 4, we present the achieved power for comparisons that were not statistically significant, including anthropometric, body composition, strength, and biochemical biomarker variables.

Supplementary table 4: Post-hoc power analysis for not-significant changes after one year of follow-up

n

Cohen’s d for paired samples

Power obtained

WC (cm)

61

0.23

0.424

WHtR (cm/cm)

61

0.22

0.394

Rz (Ohm)

61

0.14

0.190

FFMI (kg/m2)

61

0.03

0.056

ASMMI (kg/m2)

61

0.11

0.135

VF (L)

58

0.06

0.073

HGS (kg)

61

0.31

0.664

sASAT (cm)

53

0.23

0.376

Glycemia (mg/dL)

60

0.02

0.053

HbA1c (%)

47

0.23

0.339

Insulin (µUI/mL)

49

0.14

0.161

HOMA-IR

37

0.10

0.091

Total cholesterol (mg/dL)

56

0.22

0.366

HDL-cholesterol (mg/dL)

55

0.13

0.157

Non-HDL-cholesterol (mg/dL)

55

0.19

0.283

LDL-cholesterol (mg/dL)

55

0.22

0.361

Triglycerides (mg/dL)

56

0.05

0.066

CRP (mg/L)

51

0.21

0.313

IL-6 (pg/mL)

51

0.06

0.070

Vitamin D (ng/mL)

53

0.23

0.376

Myostatin (ng/mL)

41

~0.00

0.050

FGF-21 (pg/mL)

40

0.22

0,274

ASMMI: Appendicular Skeletal Muscle Mass Index; CRP: C-reactive protein; FGF-21: Fibroblast growth factor 21; FFMI: Fat-Free Mass Index; Glyc: Glycemia; HbA1c: glycosylated haemoglobin; HDL: High-density lipoprotein; HGS: Handgrip strength; HOMA-IR: Insulin Resistance Homeostatic Model Assessment; IL-6: Interleukin 6; LDL: Low density lipoproteins; Rz: Resistance; sASAT: superficial abdominal subcutaneous adipose tissue; VF: Visceral Fat; WC: Waist Cir-cumference; WHtR: Waist–to–Height Ratio

Cohen's d: difference between two means divided by the pooled standard deviation (effect size: 0.2 = small effect; 0.5 = medium effect; 0.8 = large effect). Power was calculated with jPower Module from jamovi Cloud (https://cloud.jamovi.org/), assuming a two-sided criterion for detection that allows for a maximum Type I error rate of α = 0.05.

This table presents the results of post hoc power analyses conducted for variables in which changes after one year of follow-up were not statistically significant. For each variable, the observed effect size, sample size, and corresponding statistical power are listed to provide context regarding the study's ability to detect meaningful differences over time. These analyses help clarify that the non-significant findings may be due to insufficient statistical power.

Regarding Spearman correlation analysis for absolute changes, most correlation coefficients were far below 0.500, except for ΔFM (%) vs ΔHOMA-IR (ρ = 0.522, p = 0.001), ΔFMI vs ΔHOMA-IR (ρ = 0.481, p = 0.003), and ΔFM/FFM ratio vs ΔHOMA-IR (ρ = 0.538, p = 0.001). For relative changes (%), the highest levels of rho coefficient were around 0.5 such as Δ%FM/FFM ratio vs Δ%HOMA -ρ = 0.480, p=0.003. Spearman’s rho coefficients below 0.4 generally indicate a low effect size, which reduces statistical power to detect associations given our limited sample size. Consequently, we were able to detect statistically significant associations in variables with larger effect sizes but not in weaker ones. Correlations with rho values greater than 0.35 and sample size around 50 attained statistical power in the range of 0.75 to 0.80, which is considered reasonable for detecting effects. In contrast, correlations with rho values between 0.15 and 0.25 and sample sizes under 50 had power below 0.50, indicating the analyses were underpowered. The strongest correlations (rho >0.5) with sample sizes exceeding 40 achieved power levels above 0.85, indicating robust findings (https://cloud.jamovi.org/a727f1b2-2974-42c8-a435-bc7269973fa4/; data not shown).

The following text has been included in the manuscript:

-Methods,

“As the main project focuses on the evolution of adiposity over time following breast cancer diagnosis, the sample size calculation for this exploratory analysis was based on detecting changes in the most clinically accessible adiposity marker, %FM estimated by BIA. Accordingly, a sample size of 44 paired observations would be required to reliably detect an effect size of δ ≥ 0.5 with a statistical power greater than 0.9, assuming a two-sided test and a maximum Type I error rate of α = 0.05. For US comparisons, the sample size estimate was more conservative due to the lack of established data on minimally clinically relevant changes. In this context, 52 paired observations would be needed to detect an effect size of δ ≥ 0.4 with a power greater than 0.8, also assuming a two-sided test and α = 0.05.”

-Limitations,

“The supplementary table 4 presents the results of post hoc power analyses for variables (body composition, strength and biochemical biomarkers) in which changes after one year of follow-up were not statistically significant. These analyses indicate that the non-significant findings may be attributable to insufficient statistical power, which ranged from 0.050 to 0.664. An increased sample size would be required to detect statistically significant differences. Moderate to large correlations (±0.35 and above) generally achieved acceptable power (>0.70), whereas weaker correlations (<0.30) were underpowered. Therefore, the lack of significance in correlations with low rho values (approximately 0.10–0.20) may also be due to insufficient power. In contrast, significant correlations with moderate to large rho values were adequately powered.”

  1. The broad age range (40–80 years) introduces potential confounding, as BC alterations (e.g., sarcopenia, fat redistribution) may differ substantially between younger and older patients. Stratified analyses by age group could clarify these effects.

Thank you very much for your suggestion. We agree that a stratified analysis by menopausal status, received treatment, and tumor type would add great value to the study. However, after evaluating the required sample size to detect one of the strongest associations, we found that the current sample is insufficient to support such subgroup analyses: “As the main project focuses on the evolution of adiposity over time following breast cancer diagnosis, the sample size calculation for this exploratory analysis was based on detecting changes in the most clinically accessible adiposity marker, %FM estimated by BIA. Accordingly, a sample size of 44 paired observations would be required to reliably detect an effect size of δ ≥ 0.5 with a statistical power greater than 0.9, assuming a two-sided test and a maximum Type I error rate of α = 0.05. For US comparisons, the sample size estimate was more conservative due to the lack of established data on minimally clinically relevant changes. In this context, 52 paired observations would be needed to detect an effect size of δ ≥ 0.4 with a power greater than 0.8, also assuming a two-sided test and α = 0.05.”). The potential advantages may be outweighed by the drawbacks or limitations, given the limited number of pairs available in each subgroup (premenopausal/postmenopausal: 16/45;radiotherapy/non-radiotherapy: 45/16; chemotherapy/non-chemotherapy: 14/47; hormonal therapy/non-hormonal therapy: 52/9; luminal A and B/other types: 41/20). This limitation has been explained in the limitations section of the manuscript:

“Therefore, this sample size has not allowed us to perform valid sub-analyses based on menopausal status or the treatment received, which have been identified as risk factors for deleterious changes in BC [34].”

In contrast, we conducted multiple regression analyses adjusting for age on the significant associations identified in the bivariate correlation analysis. This topic has been thoroughly explained both in this cover letter and in the manuscript:

Several variables may act as confounders or effect modifiers, but the sample size is not large enough to allow for proper adjustment. We have now explored the significant associations identified in the correlation analysis to determine possible adjustments, using rank transformation of the data since not all variables satisfied the assumption of normality. Among the variables considered, only age was not affected by collinearity. The crude and age-adjusted models are presented in Supplementary Table 3.

Supplementary table 3: Multiple linear regression models used to explain the relationship between changes in body composition parameters (independent variable) and the evolution of biomarkers (dependent variable): crude models and adjusted for age.

Associated rank-transformed variables

Standardized β; p

DWeight - DHDL-cholesterol

Crude:              -0.319; 0.018

Adjusted:         -0.298; 0.030

DWeight - DVitamin D

Crude:              -0.339; 0.013

Adjusted:         -0.356; 0.011

DWeight - DMyostatin

Crude:              -0.360; 0.021

Adjusted:         -0.348; 0.027

DBMI - DHDL-cholesterol

Crude:              -0.289; 0.034

Adjusted:         -0.296; 0.030

DBMI - DTriglycerides

Crude:              0.272; 0.043

Adjusted:         0.302; 0.028

DBMI - DVitamin D

Crude:              -0.308; 0.025

Adjusted:         -0.322; 0.022

DBMI - DMyostatin

Crude:              -0.360; 0.021

Adjusted:         -0.348; 0.027

DWC  -DIL-6

Crude:              -0.322; 0.021

Adjusted:         -0.318; 0.024

DWC/Ht-DIL-6

Crude:              -0.317; 0.024

Adjusted:         -0.313; 0.027

DPhA/BMI  - DHbA1c

Crude:              0.386; 0.007

Adjusted:         0.415; 0.004

DECW/TBW -DHbA1c

Crude:              -0.311; 0.033

Adjusted:         -0.314; 0.032

DFM  -DInsulin

Crude:              0.330; 0.020

Adjusted:         0.303; 0.039

DFM - DHOMA-IR

Crude:              0.503; 0.002

Adjusted:         0.466; 0.006

DFM - DInsulin

Crude:              0.327; 0.022

Adjusted:         0.302; 0.037

DFM  - DHOMA-IR

Crude:              0.436; 0.017

Adjusted:         0.390; 0.020

DFMI  -DInsulin

Crude:              0.341; 0.017

Adjusted:         0.316; 0.030

DFMI - DHOMA-IR

Crude:              0.461; 0.004

Adjusted:         0.417; 0.013

DVF  - DCRP

Crude:              0.305; 0.035

Adjusted:         0.308; 0.035

DThigh SAT  -DInsulin

Crude:              0.349; 0.025

Adjusted:         0.324; 0.035

DtASAT  - DHDL-cholesterol

Crude:              -0.447; 0.002

Adjusted:         -0.444; 0.002

DtASAT - DTriglycerides

Crude:              0.381; 0.008

Adjusted:         0.382; 0.008

DsASAT  - DHDL-cholesterol

Crude:              -0.398; 0.006

Adjusted:         -0.395; 0.005

DsASAT - DTriglycerides

Crude:              0.405; 0.004

Adjusted:         0.405; 0.005

DsASAT  - DIL-6

Crude:              0.340; 0.024

Adjusted:         0.340; 0.026

DPreperitonealAT -DVitamin D

Crude:              -0.298; 0.046

Adjusted:         -0.299; 0.049

DFFMI  -DHOMA-IR

Crude:              -0.333; 0.044

Adjusted:         -0.328; 0.042

DSMMI - DTotal-cholesterol

Crude:              -0.261; 0.052

Adjusted:         -0.255; 0.053

DSMMI - Dnon-HDL-cholesterol

Crude:              -0.260; 0.055

Adjusted:         -0.260; 0.053

DASMMI  -DHDL-cholesterol

Crude:              -0.279; 0.039

Adjusted:         -0.263; 0.053

DASMMI  -DVitamin D

Crude:              -0.303; 0.027

Adjusted:         -0.309; 0.026

 D RF thickness - DMyostatin

Crude:              -0.288; 0.084

Adjusted:         -0.288; 0.087

DFM/FFM ratio -DInsulin

Crude:              0.358; 0.012

Adjusted:         0.335; 0.020

DFM/FFM ratio -DHOMA-IR

Crude:              0.524; 0.001

Adjusted:         0.487; 0.003

DSMM/weight - DHbA1c

Crude:              0.414; 0.004

Adjusted:         0.477; 0.001

DSMM/weight  - DHOMA-IR

Crude:              -0.335; 0.042

Adjusted:         -0.286; 0.089

DSMM/weight -DTotal-cholesterol

Crude:              -0.268; 0.046

Adjusted:         -0.326; 0.014

DASMM/BMI  -DHDL-cholesterol

Crude:              0.274; 0.043

Adjusted:         0.253; 0.065

DASMM/BMI  -Dnon-HDL-cholesterol

Crude:              -0.275; 0.042

Adjusted:         -0.316; 0.019

DASMM/BMI  -DCRP

Crude:              -0.352; 0.011

Adjusted:         -0.344; 0.015

D%Weight -D%HDL-cholesterol

Crude:              -0.338; 0.012

Adjusted:         -0.319; 0.019

D%Weight - D%Vitamin D

Crude:              -0.373; 0.006

Adjusted:         -0.389; 0.005

D%Weight - D%Myostatin

Crude:              -0.335; 0.033

Adjusted:         -0.321; 0.041

D%BMI - D%HDL-cholesterol

Crude:              -0.330; 0.014

Adjusted:         -0.310; 0.022

D%BMI - D% Triglycerides

Crude:              0.268; 0.046

Adjusted:        0.290; 0.034

D%BMI - D%Vitamin D

Crude:              -0.330; 0.016

Adjusted:         -0.344; 0.014

D%BMI - D%Myostatin

Crude:              -0.355; 0.023

Adjusted:         -0.342; 0.029

D%WC - D%IL-6

Crude:              -0.326; 0.019

Adjusted:        -0.325; 0.022

D%WC/Ht -  D%IL-6

Crude:              -0.326; 0.019

Adjusted:        -0.325; 0.022

D%Xc - D% Triglycerides

Crude:              0.284; 0.034

Adjusted:        0.284; 0.035

D%PhA/BMI - D%HbA1c

Crude:              0.397; 0.006

Adjusted:        0.427; 0.003

D%FM_% - D%HbA1c

Crude:             -0.324; 0.026

Adjusted:       -0.450; 0.003

D%FM_% - D%Insulin

Crude:              0.278; 0.053

Adjusted:        0.241; 0.104

D%FM_% -D%HOMA

Crude:              0.452; 0.005

Adjusted:        0.400; 0.020

D%FM_% -D%HDL-cholesterol

Crude:              -0.282; 0.037

Adjusted:         -0.256; 0.069

D%FM_kg - D%HbA1c

Crude:             -0.295; 0.044

Adjusted:       -0.391; 0.010

D%FM_kg -D%HOMA

Crude:              0.362; 0.028

Adjusted:        0.301; 0.077

D%FM_kg -D%HDL-cholesterol

Crude:              -0.288; 0.033

Adjusted:         -0.264; 0.057

D%FM_kg - D%CRP

Crude:              0.306; 0.029

Adjusted:        0.297; 0.039

D%FMI  -D%HOMA

Crude:              0.430; 0.008

Adjusted:        0.375; 0.026

D%FMI -D%HDL-cholesterol

Crude:              -0.273; 0.044

Adjusted:         -0.246; 0.078

D%FMI - D%CRP

Crude:              0.237; 0.094

Adjusted:        0.226; 0.120

D%Thigh SAT -D%Insulin

Crude:              0.362; 0.020

Adjusted:        0.346; 0.022

D%tASAT - D%HDL-cholesterol

Crude:              -0.460; 0.001

Adjusted:         -0.456; 0.001

D%tASAT - D% Triglycerides

Crude:              0.329; 0.023

Adjusted:         0.329; 0.024

D%sASAT - D%HDL-cholesterol

Crude:              -0.410; 0.004

Adjusted:         -0.408; 0.004

D%sASAT - D% Triglycerides

Crude:              0.368; 0.010

Adjusted:         0.368; 0.011

D%PreperitonealAT - D%Vitamin D

Crude:              -0.293; 0.051

Adjusted:         -0.297; 0.052

D%FFMI - D%HOMA

Crude:              -0.351; 0.033

Adjusted:        -0.343; 0.032

D%ASMMI -D%HDL-cholesterol

Crude:              -0.278; 0.040

Adjusted:         -0.263; 0.053

D%ASMMI -D%Vitamin D

Crude:              -0.339; 0.013

Adjusted:         -0.345; 0.013

D%RF-CSA - D%CRP

Crude:              -0.325; 0.036

Adjusted:         -0.316; 0.045

D%RF-CSA -D%Vitamin D

Crude:              0.309; 0.041

Adjusted:         0.317; 0.038

D%SMM/weight  - D%HbA1c

Crude:              0.460; 0.001

Adjusted:         0.513; <0.001

D%SMM/weight - D%HOMA

Crude:              -0.372; 0.023

Adjusted:         -0.327; 0.046

D%ASMM/BMI -D%HbA1c

Crude:              0.315; 0.031

Adjusted:         0.379; 0.011

D%ASMM/BMI -D%HDL-cholesterol

Crude:              0.284; 0.035

Adjusted:         0.265; 0.052

D%ASMM/BMI -D%non-HDL-cholesterol

Crude:              -0.264; 0.051

Adjusted:         -0.298; 0.028

D%ASMM/BMI -D%CRP

Crude:              -0.340; 0.015

Adjusted:         -0.334; 0.018

The variables were rank transformed to be included in the models. ASMM: Appendicular Skeletal Muscle Mass; ASMMI: Appendicular Skeletal Muscle Mass Index; AT: Adipose Tissue; BMI: Body Mass Index; CRP: C-reactive protein; ECW: Extracellular Water; FFM: Fat-Free Mass; FFMI: Fat-Free Mass Index; FM: Fat Mass; FMI: Fat Mass Index; HbA1c: glycosylated haemoglobin; HDL: High-density lipoprotein; HOMA-IR: Insulin Resistance Homeostatic Model Assessment; IL-6: Interleukin 6; PhA: Phase Angle; RF: rectus femoris; RF-CSA: rectus femoris cross-sectional area; SAT: subcutaneous adipose tissue; sASAT: superficial abdominal subcutaneous adipose tissue; SMM: Skeletal Muscle Mass; SMMI: Skeletal Muscle Mass  index; tASAT: total abdominal subcutaneous adipose tissue; TBW: Total Body Water; VF: Visceral Fat; WC: Waist Circumference; WHtR: Waist–to–Height Ratio; Xc: Reactance.

As the Spearman rho coefficient helped us exploratorily understand the associations between changes in body composition parameters and biochemical biomarkers, this new analysis allowed us to examine the average change in ranks of the dependent variables (biochemical biomarkers) for each one-rank increase in the studied body composition parameters (independent variables), adjusting for age.

When examining absolute differences, almost all relationships remained statistically significant and retained the same direction of association, as reflected by the standardized beta coefficients after adjustment. However, statistical significance was lost after adjustment in the following pairs: DASMMI (kg/m2) -DHDL-cholesterol (mg/dL), DSMM/weight (%) - DHOMA-IR, and DASMM/BMI (kg/kg*m2) -DHDL-cholesterol (mg/dL).

In the univariate analysis, associations between  DSMMI (kg/m2) - DTotal-cholesterol (mg/dL), DSMMI (kg/m2) - Dnon-HDL-cholesterol (mg/dL), and D RF thickness (cm) - DMyostatin (ng/mL) were not observed, suggesting that the correlations previously detected may represent more spurious associations.

When relative (%) differences where explored, age acted as a confounder in the following pairs: D%FM_% -D%HDL-cholesterol, D%FM_kg -D%HOMA, D%FM_kg -D%HDL-cholesterol, D%FMI -D%HDL-cholesterol, D%ASMMI -D%HDL-cholesterol, D%ASMM/BMI (kg/kg*m2) -D%non-HDL-cholesterol, and D%ASMM/BMI (kg/kg*m2) -D%HDL-cholesterol.

No significant associations were found in the univariate analysis for these pairs: D%FM_% - D%Insulin, D%FMI - D%CRP, and D%PreperitonealAT - D%Vitamin D.

We have added these paragraphs to the manuscript:

  1. Statistical section: “To account for confounders in the associations between changes in body composition parameters and biochemical biomarkers, we performed both crude and age-adjusted linear regression analyses using rank-transformed variables that showed statistically significant associations in the correlation analysis. Since the body composition parameters (independent variables) were interrelated and given the limited sample size, only age (rank-transformed) was included as a covariate.”
  2. Results section:

b.1. Title: “Association” (instead of Correlations) between body composition changes and biomarkers evolution

b.2. “The crude and age-adjusted linear regression models are presented in the supplementary table 3. Robust and biologically plausible associations were observed between body composition parameters and key metabolic biomarkers after adjustment for age. Increased FM, as measured by DFM (%), D%FM_% DFMI, DFM/FFM ratio, showed strong positive associations with DHOMA-IR and D%HOMA (adjusted β ranging from 0.400 to 0.487; p<0.025). Increased SAT, as measured by DtASAT, D%tASAT and D%sASAT demonstrated strong negative associations with DHDL-cholesterol and D%HDL-cholesterol (adjusted β ranging from 0.408 to 0.456; p<0.005). In addition, DsASAT was significantly and positively associated with DTriglycerides (adjusted β  0.405; p=0.005). However, some of the studied associations were no longer significant after adjusting for age.” 

  1. Discussion section:

c.1. “Our analyses revealed several robust and statistically significant associations between changes in body composition, classical anthropometric parameters and biochemical biomarkers, independent of age. Notably, increases in FM—expressed as percentage FM and FMI—were strongly and positively associated with markers of insulin resistance, including HOMA-IR. Conversely, SAT in thigh and abdominal regions showed consistent inverse relationships with HDL-cholesterol and positive associations with triglycerides, highlighting their impact on lipid metabolism.“

c.2. “The results of both crude and age-adjusted linear regression models revealed several statistically significant associations between changes in body composition, classical anthropometric parameters, and biochemical biomarkers. However, some of these associations were no longer significant after adjusting for age. This attenuation of significance after adjustment suggests that age influences these associations and may partly explain the correlations observed in the crude analyses.”

  1. The absence of a non-cancer control group makes it difficult to discern whether observed BC changes are specific to breast cancer treatment or reflect age-related metabolic shifts.

We agree with the reviewer that the absence of a control group is a limitation of our study, and we have now explicitly acknowledged this in the revised manuscript. However, preliminary studies have reported significant differences in body composition between breast cancer survivors and women without a cancer history, suggesting that cancer and its treatments may play a role in these changes. Although we were unable to include a matched control group in this initial phase, our ongoing study includes a long-term follow-up with an expanded cohort, which will allow for stratified analyses and better contextualization of these findings (Prevalence of sarcopenia in women with breast cancer. Nutrients 2022, 14, 1839. https://doi.org/10.3390/nu14091839 ).

This sentence was included in the limitations section:

The lack of a control group is another limitation of our study; however, preliminary reports suggest that cancer and its treatments may contribute to the observed changes in body composition.”

Morlino D, Marra M, Cioffi I, Santarpia L, De Placido P, Giuliano M, et al. Prevalence of Sarcopenia in Women with Breast Cancer. Nutrients. 2022; 14(9):1839. doi: 10.3390/nu14091839. PMID: 35565806; PMCID: PMC9099516.

  1. In this study, authors did not measure Leptin/adiponectin or IGF-1, which could provide mechanistic insights into fat redistribution effects.

We appreciate your valuable comment regarding the absence of leptin, adiponectin, and IGF-1 measurements in our study. We acknowledge that these biomarkers provide important insights into the endocrine regulation of body composition. Similar to leptin and adiponectin—which we plan to include in future analyses as our laboratory has the necessary resources—FGF1 is currently beyond our capabilities. In our initial research, we focused on FGF21 as a novel biomarker with potential to link systemic metabolic status, body mass redistribution, and pathological processes such as cancer. Its role as an intermediary between metabolism and cellular function makes it an attractive candidate for understanding how interventions that modify body composition might influence the biological environment related to cancer, beyond simple energy balance. However, in our current sample, we have not yet achieved sufficient statistical power to confirm or refute its role in breast cancer.

  1. While BIA is practical in this study, it is less accurate than DEXA or CT/MRI for body composition assessment, particularly in cancer patients with fluid shifts.

We agree with your point and have addressed it with specific comments in the manuscript.

Many BIA-derived parameters are calculated using formulas developed in specific healthy populations, which may not apply to ill individuals whose water distribution may differ from the standard. Moreover, these formulas are particularly inaccurate at the extremes of BMI.

However, the most recent ASPEN guidelines¹ state: “BIA is a practical, portable, non-invasive tool that poses minimal risks and low costs relative to the other body composition assessment methods. These characteristics allow its use in any setting, such as epidemiological and clinical studies, and render BIA an ideal assessment technique for follow-up studies, in which repeated measurements are necessary and easily obtained. (…) An acceptable mean level accuracy for BIA assessments has been established in healthy, nonobese participants.”

Therefore, in a clinical study such as this, BIA may provide an opportunity to move beyond the simplistic measurement of BMI. Raw parameters such as phase angle, resistance, and reactance should be described as markers of cellular status and their relationship with hydration. However, in this exploratory study, associations between these parameters and changes in biochemical biomarkers were not generally observed. Some associations were found with changes in body compartments, such as body fat and skeletal muscle mass, when corrected by adiposity parameters such as SMM/weight or ASMM/BMI. This supports biological plausibility and, to a certain extent, validates the use of this method of body composition assessment in the study.

Furthermore, the volunteers enrolled at baseline were women with recently diagnosed breast cancer not yet affected by cancer treatment. Although the study included older adults, only 8.2% (n=5) of the sample were aged 75 years or older. The BMI range at baseline was 15.4 to 38.6 kg/m², and 5 individuals (8.2%) had BMI values below 18.5 or above 34 kg/m². At the end of follow-up, 9.8% (n=6) were over 75 years old, and 6.6% (n=4) were at the extremes of BMI. It is possible that BIA parameters were less accurate in these subgroups. Additionally, overestimation of FM and SMM may have occurred after one year of therapy, considering that increases in body water are commonly seen with disease progression, as noted in the discussion section: “This increase in TBW represents a limitation of BIA, as it may lead to an underestimation of FM and an overestimation of FFM in these patients.” In any case, our study did not show changes in crude data related to FFM and ASMMI, whereas changes were observed in FM-related metrics, thus allowing BIA analysis to differentiate between body compartments not described by anthropometric measurements.

Of course, CT or MRI could provide more precise tools for body composition assessment. However, CT is not always available in breast cancer follow-up, and MRI is not commonly used in the clinical management of this disease. We are currently studying CT-based muscle and fat changes in those cases where this approach is possible, and we hope to present reliable data in the future.

We have added the following statement to the limitations section: “On the other hand, BIA analysis may not be accurate enough to estimate body compartments in patients at the extremes of age or BMI, or with altered hydration status. Nevertheless, the changes observed during follow-up were consistent with previously published studies [44], and although an increase in ECW/TBW was observed, the percentage and absolute values of TBW did not change in a clinically relevant manner (from 28.33 ± 3.82 L to 28.40 ± 3.63 L, and from 45.69 ± 4.92% to 44.75 ± 4.74%). More studies are necessary to assess the external validity of BIA-derived data in breast cancer.”

1.- Sheean P, Gonzalez MC, Prado CM, McKeever L, Hall AM, Braunschweig CA. American Society for Parenteral and Enteral Nutrition Clinical Guidelines: The Validity of Body Composition Assessment in Clinical Populations. JPEN J Parenter Enteral Nutr. 2020 Jan;44(1):12-43. doi: 10.1002/jpen.1669. Epub 2019 Jun 19. PMID: 31216070.

  1. While fat parameters correlated with insulin resistance and lipids, the study cannot determine directionality. Does fat gain drive dysregulation, or vice versa?

At this point, and given the relatively short follow-up period of our study, it is not possible to establish a directional link between body composition changes and metabolic derangements in breast cancer. Any comment on this relationship would be speculative. However, our data show that changes in body composition occur during the first year after diagnosis, yet these are not yet reflected in alterations of biochemical variables. Therefore, we might suggest that alterations in body composition precede metabolic changes. We have included this statement in the future research section:

“Further investigations should also explore the predictive value of early changes in body composition for clinical outcomes such as cardiovascular events, frailty, and reduced quality of life”.

  1. Authors measured handgrip strength (Jamar dynamometer), which may not fully reflect lower-body or axial muscle function (critical for mobility and metabolic health).

This is an important point that we have examined in our sample; however, it appeared to be poorly discriminative in a preliminary overview, likely due to the age and health status of the included women. We used the chair stand test to assess muscle strength and, to some extent, muscle function. During the test, patients repeatedly stood up fully from a seated position and then sat back down. We measured the time, in seconds, required to complete five full sit-to-stand cycles:

Basal

One year

Absolute change

Relative change (%)

p-value

Sit-to-stand (sec.)

10.50

(8.69; 13.30)

10.49

(8.62; 1.30)

-0.30

(-2.56; 2.04)

-0.23

(-20.32; 21.42)

0.923

 There were no differences in test performance after one year of follow-up. Low performance, defined as 15 seconds or more to complete the test, was observed in 9 patients (15.2%) at baseline and in 10 patients (16.9%) at one-year post-diagnosis.

  1. Did patients receive chemotherapy, radiation, or endocrine therapy? For instance, aromatase inhibitors are known to exacerbate muscle loss.

This is, in fact, a significant aspect that will be assessed in future research involving our cohort. The relevant data from the studied sample are presented in Table 1.

Total (n=61)

Premenopausal status (n=16)

Menopausal status (n=45)

p-value

Treatment (after surgery)

ChT

14 (23.0)

4 (25.0)

10 (22.2)

1.000

RT

45 (73.8)

14 (87.5)

31 (68.9)

0.196

HT

52 (85.3)

14 (87.5)

38 (84.4)

1.000

Tamoxifen

22 (36.1)

12 (75.0)

10 (22.2)

0.014

Letrozole

22 (36.1)

2 (12.5)

20 (44.4)

Anastrozole

8 (13.1)

0 (0.0)

8 (17.8)

ChT: Chemotherapy; HT: Hormone therapy; RT: Radiation therapy

Nevertheless, the sample size does not permit stratified analysis as previously mentioned, due to the number of pairs required to achieve adequate statistical power.

  1. HOMA-IR is derived from fasting glucose/insulin, which may not capture postprandial dysregulation (common in metabolic syndrome).

We appreciate your observation regarding the limitations of HOMA-IR in capturing postprandial glucose-insulin dysregulation. However, we believe that HOMA-IR remains a valid and clinically useful tool for detecting metabolic alteration, even when fasting glucose and HbA1c are within normal ranges.

For example, the study by Onat et al. demonstrated that a significant proportion of adults with normal glucose levels exhibited a discordance between metabolic syndrome and insulin resistance, with a high prevalence of elevated HOMA-IR in individuals without metabolic syndrome. Even in the absence of postprandial glycemic alterations, these individuals showed increased cardiovascular risk, suggesting that HOMA-IR can identify subclinical and clinically meaningful metabolic dysfunction (Onat A, et al. Discordance between insulin resistance and metabolic syndrome: features and associated cardiovascular risk in adults with normal glucose regulation. Metabolism. 2006;55(4):445-52. doi: 10.1016/j.metabol.2005.10.005. PMID: 16546474). Iglesias-Grau et al. found that insulin resistance measured by HOMA-IR was associated with subclinical atherosclerosis in normoglycemic, low-risk individuals. This further supports the role of HOMA-IR as a sensitive early marker of metabolically relevant alteration (Iglesies-Grau J, et al. Early insulin resistance in normoglycemic low-risk individuals is associated with subclinical atherosclerosis. Cardiovasc Diabetol. 2023;22(1):350. doi: 10.1186/s12933-023-02090-1. PMID: 38115031; PMCID: PMC10731750).

In another study, elevated HOMA-IR was an early predictor of incident type 2 diabetes and chronic kidney disease, independently of HbA1c, in non-diabetic individuals (Lee J, et al. Assessment HOMA as a predictor for new onset diabetes mellitus and diabetic complications in non-diabetic adults: a KoGES prospective cohort study. Clin Diabetes Endocrinol. 2023; 9(1):7. doi: 10.1186/s40842-023-00156-3. PMID: 37974292; PMCID: PMC10652621.)

Therefore, we believe that combining HOMA-IR and HbA1c provides a more comprehensive picture and helps to detect subclinical metabolic alterations in early stages, before sustained hyperglycemia becomes apparent.

Reviewer 3 Report

Comments and Suggestions for Authors

I read with interest the manuscript “Body composition changes and their relationship with biomarkers evolution one year after breast cancer diagnosis”.

The study explores a clinically relevant but already extensively discussed topic—body composition changes post-breast cancer diagnosis. The novelty lies in the combined use of BIA and ultrasonography (US), but the manuscript lacks depth in the interpretation of these methods’ comparative value and limitations.

Major concern

  1. The methodology is comprehensive but lacks clarity in several areas:

Sample Size: The final sample (n=61) is relatively small and weakens statistical power, especially when subgroup analysis (e.g., menopausal status) is hinted at but not performed.

Inclusion Criteria: More detail is required on recruitment, particularly regarding the impact of ongoing treatments on BC and biomarker evolution.

Control Group: The absence of a control group significantly limits causal inferences.

Operator Variability: The US was performed by two individuals, which could introduce bias, yet inter-rater reliability is neither assessed nor mentioned.

  1. Statistical analysis: Several p-values are close to the significance threshold and should be interpreted cautiously. No adjustments for multiple comparisons are discussed, which increases the risk of Type I error. The rationale for choosing specific correlation coefficients and the implications of non-significant results are insufficiently discussed.
  2. While changes in fat distribution and muscle parameters are well documented, the lack of significant changes in biomarkers undermines the strength of the manuscript’s key hypothesis. The rise in sarcopenic obesity prevalence (from 1.16% to 4.9%) is statistically minor and clinically questionable without stronger supporting data.
  3. The discussion occasionally overstates conclusions from non-significant findings. More attention should be paid to the influence of confounding variables (e.g., treatment regimens, physical activity, nutritional counseling). The link between muscle deconditioning and functional prognosis needs stronger justification, especially given that HGS remained unchanged.
  4. The manuscript is readable but includes frequent grammatical errors, unclear sentence structures, and inconsistent terminology (e.g., "BC" for body composition is not universally recognized and should be standardized).

Author Response

Reviewer 3:

Comments and Suggestions for Authors

I read with interest the manuscript “Body composition changes and their relationship with biomarkers evolution one year after breast cancer diagnosis”.

The study explores a clinically relevant but already extensively discussed topic—body composition changes post-breast cancer diagnosis. The novelty lies in the combined use of BIA and ultrasonography (US), but the manuscript lacks depth in the interpretation of these methods’ comparative value and limitations.

1) Sample Size: The final sample (n=61) is relatively small and weakens statistical power, especially when subgroup analysis (e.g., menopausal status) is hinted at but not performed.

Thank you for reading the manuscript and made valuable contributions for the improvement of the paper.

Regarding sample size, we have thoroughly reviewed the data to calculate the statistical power achieved in non-significant associations and to discuss the limitations of performing stratified analyses. Recognizing age as a potentially important confounding variable, we conducted adjusted analyses controlling for age on the significant associations identified in the bivariate correlation analysis. We detail and explain these considerations below:

In Supplementary Table 4, we present the achieved power for comparisons that were not statistically significant, including anthropometric, body composition, strength, and biochemical biomarker variables.

Supplementary table 4: Post-hoc power analysis for not-significant changes after one year of follow-up

n

Cohen’s d for paired samples

Power obtained

WC (cm)

61

0.23

0.424

WHtR (cm/cm)

61

0.22

0.394

Rz (Ohm)

61

0.14

0.190

FFMI (kg/m2)

61

0.03

0.056

ASMMI (kg/m2)

61

0.11

0.135

VF (L)

58

0.06

0.073

HGS (kg)

61

0.31

0.664

sASAT (cm)

53

0.23

0.376

Glycemia (mg/dL)

60

0.02

0.053

HbA1c (%)

47

0.23

0.339

Insulin (µUI/mL)

49

0.14

0.161

HOMA-IR

37

0.10

0.091

Total cholesterol (mg/dL)

56

0.22

0.366

HDL-cholesterol (mg/dL)

55

0.13

0.157

Non-HDL-cholesterol (mg/dL)

55

0.19

0.283

LDL-cholesterol (mg/dL)

55

0.22

0.361

Triglycerides (mg/dL)

56

0.05

0.066

CRP (mg/L)

51

0.21

0.313

IL-6 (pg/mL)

51

0.06

0.070

Vitamin D (ng/mL)

53

0.23

0.376

Myostatin (ng/mL)

41

~0.00

0.050

FGF-21 (pg/mL)

40

0.22

0,274

ASMMI: Appendicular Skeletal Muscle Mass Index; CRP: C-reactive protein; FGF-21: Fibroblast growth factor 21; FFMI: Fat-Free Mass Index; Glyc: Glycemia; HbA1c: glycosylated haemoglobin; HDL: High-density lipoprotein; HGS: Handgrip strength; HOMA-IR: Insulin Resistance Homeostatic Model Assessment; IL-6: Interleukin 6; LDL: Low density lipoproteins; Rz: Resistance; sASAT: superficial abdominal subcutaneous adipose tissue; VF: Visceral Fat; WC: Waist Cir-cumference; WHtR: Waist–to–Height Ratio

Cohen's d: difference between two means divided by the pooled standard deviation (effect size: 0.2 = small effect; 0.5 = medium effect; 0.8 = large effect). Power was calculated with jPower Module from jamovi Cloud (https://cloud.jamovi.org/), assuming a two-sided criterion for detection that allows for a maximum Type I error rate of α = 0.05.

This table presents the results of post hoc power analyses conducted for variables in which changes after one year of follow-up were not statistically significant. For each variable, the observed effect size, sample size, and corresponding statistical power are listed to provide context regarding the study's ability to detect meaningful differences over time. These analyses help clarify that the non-significant findings may be due to insufficient statistical power.

Regarding Spearman correlation analysis for absolute changes, most correlation coefficients were far below 0.500, except for ΔFM (%) vs ΔHOMA-IR (ρ = 0.522, p = 0.001), ΔFMI vs ΔHOMA-IR (ρ = 0.481, p = 0.003), and ΔFM/FFM ratio vs ΔHOMA-IR (ρ = 0.538, p = 0.001). For relative changes (%), the highest levels of rho coefficient were around 0.5 such as Δ%FM/FFM ratio vs Δ%HOMA -ρ = 0.480, p=0.003. Spearman’s rho coefficients below 0.4 generally indicate a low effect size, which reduces statistical power to detect associations given our limited sample size. Consequently, we were able to detect statistically significant associations in variables with larger effect sizes but not in weaker ones. Correlations with rho values greater than 0.35 and sample size around 50 attained statistical power in the range of 0.75 to 0.80, which is considered reasonable for detecting effects. In contrast, correlations with rho values between 0.15 and 0.25 and sample sizes under 50 had power below 0.50, indicating the analyses were underpowered. The strongest correlations (rho >0.5) with sample sizes exceeding 40 achieved power levels above 0.85, indicating robust findings (https://cloud.jamovi.org/a727f1b2-2974-42c8-a435-bc7269973fa4/; data not shown).

The following text has been included in the manuscript:

-Methods,

“As the main project focuses on the evolution of adiposity over time following breast cancer diagnosis, the sample size calculation for this exploratory analysis was based on detecting changes in the most clinically accessible adiposity marker, %FM estimated by BIA. Accordingly, a sample size of 44 paired observations would be required to reliably detect an effect size of δ ≥ 0.5 with a statistical power greater than 0.9, assuming a two-sided test and a maximum Type I error rate of α = 0.05. For US comparisons, the sample size estimate was more conservative due to the lack of established data on minimally clinically relevant changes. In this context, 52 paired observations would be needed to detect an effect size of δ ≥ 0.4 with a power greater than 0.8, also assuming a two-sided test and α = 0.05.”

-Limitations,

“The supplementary table 4 presents the results of post hoc power analyses for variables (body composition, strength and biochemical biomarkers) in which changes after one year of follow-up were not statistically significant. These analyses indicate that the non-significant findings may be attributable to insufficient statistical power, which ranged from 0.050 to 0.664. An increased sample size would be required to detect statistically significant differences. Moderate to large correlations (±0.35 and above) generally achieved acceptable power (>0.70), whereas weaker correlations (<0.30) were underpowered. Therefore, the lack of significance in correlations with low rho values (approximately 0.10–0.20) may also be due to insufficient power. In contrast, significant correlations with moderate to large rho values were adequately powered.”

“As the main project focuses on the evolution of adiposity over time following breast cancer diagnosis, the sample size calculation for this exploratory analysis was based on detecting changes in the most clinically accessible adiposity marker, %FM estimated by BIA. Accordingly, a sample size of 44 paired observations would be required to reliably detect an effect size of δ ≥ 0.5 with a statistical power greater than 0.9, assuming a two-sided test and a maximum Type I error rate of α = 0.05. For US comparisons, the sample size estimate was more conservative due to the lack of established data on minimally clinically relevant changes. In this context, 52 paired observations would be needed to detect an effect size of δ ≥ 0.4 with a power greater than 0.8, also assuming a two-sided test and α = 0.05.”). The potential advantages conducting stratified analysis may be outweighed by the drawbacks or limitations, given the limited number of pairs available in each subgroup (premenopausal/postmenopausal: 16/45; radiotherapy/non-radiotherapy: 45/16; chemotherapy/non-chemotherapy: 14/47; hormonal therapy/non-hormonal therapy: 52/9; luminal A and B/other types: 41/20). This limitation has been explained in the limitations section of the manuscript:

“Therefore, this sample size has not allowed us to perform valid sub-analyses based on menopausal status or the treatment received, which have been identified as risk factors for deleterious changes in BC [34].”

In contrast, we conducted multiple regression analyses adjusting for age on the significant associations identified in the bivariate correlation analysis. This topic has been thoroughly explained both in this cover letter and in the manuscript:

Several variables may act as confounders or effect modifiers, but the sample size is not large enough to allow for proper adjustment. We have now explored the significant associations identified in the correlation analysis to determine possible adjustments, using rank transformation of the data since not all variables satisfied the assumption of normality. Among the variables considered, only age was not affected by collinearity. The crude and age-adjusted models are presented in Supplementary Table 3.

Supplementary table 3: Multiple linear regression models used to explain the relationship between changes in body composition parameters (independent variable) and the evolution of biomarkers (dependent variable): crude models and adjusted for age.

Associated rank-transformed variables

Standardized β; p

DWeight - DHDL-cholesterol

Crude:              -0.319; 0.018

Adjusted:         -0.298; 0.030

DWeight - DVitamin D

Crude:              -0.339; 0.013

Adjusted:         -0.356; 0.011

DWeight - DMyostatin

Crude:              -0.360; 0.021

Adjusted:         -0.348; 0.027

DBMI - DHDL-cholesterol

Crude:              -0.289; 0.034

Adjusted:         -0.296; 0.030

DBMI - DTriglycerides

Crude:              0.272; 0.043

Adjusted:         0.302; 0.028

DBMI - DVitamin D

Crude:              -0.308; 0.025

Adjusted:         -0.322; 0.022

DBMI - DMyostatin

Crude:              -0.360; 0.021

Adjusted:         -0.348; 0.027

DWC  -DIL-6

Crude:              -0.322; 0.021

Adjusted:         -0.318; 0.024

DWC/Ht-DIL-6

Crude:              -0.317; 0.024

Adjusted:         -0.313; 0.027

DPhA/BMI  - DHbA1c

Crude:              0.386; 0.007

Adjusted:         0.415; 0.004

DECW/TBW -DHbA1c

Crude:              -0.311; 0.033

Adjusted:         -0.314; 0.032

DFM  -DInsulin

Crude:              0.330; 0.020

Adjusted:         0.303; 0.039

DFM - DHOMA-IR

Crude:              0.503; 0.002

Adjusted:         0.466; 0.006

DFM - DInsulin

Crude:              0.327; 0.022

Adjusted:         0.302; 0.037

DFM  - DHOMA-IR

Crude:              0.436; 0.017

Adjusted:         0.390; 0.020

DFMI  -DInsulin

Crude:              0.341; 0.017

Adjusted:         0.316; 0.030

DFMI - DHOMA-IR

Crude:              0.461; 0.004

Adjusted:         0.417; 0.013

DVF  - DCRP

Crude:              0.305; 0.035

Adjusted:         0.308; 0.035

DThigh SAT  -DInsulin

Crude:              0.349; 0.025

Adjusted:         0.324; 0.035

DtASAT  - DHDL-cholesterol

Crude:              -0.447; 0.002

Adjusted:         -0.444; 0.002

DtASAT - DTriglycerides

Crude:              0.381; 0.008

Adjusted:         0.382; 0.008

DsASAT  - DHDL-cholesterol

Crude:              -0.398; 0.006

Adjusted:         -0.395; 0.005

DsASAT - DTriglycerides

Crude:              0.405; 0.004

Adjusted:         0.405; 0.005

DsASAT  - DIL-6

Crude:              0.340; 0.024

Adjusted:         0.340; 0.026

DPreperitonealAT -DVitamin D

Crude:              -0.298; 0.046

Adjusted:         -0.299; 0.049

DFFMI  -DHOMA-IR

Crude:              -0.333; 0.044

Adjusted:         -0.328; 0.042

DSMMI - DTotal-cholesterol

Crude:              -0.261; 0.052

Adjusted:         -0.255; 0.053

DSMMI - Dnon-HDL-cholesterol

Crude:              -0.260; 0.055

Adjusted:         -0.260; 0.053

DASMMI  -DHDL-cholesterol

Crude:              -0.279; 0.039

Adjusted:         -0.263; 0.053

DASMMI  -DVitamin D

Crude:              -0.303; 0.027

Adjusted:         -0.309; 0.026

 D RF thickness - DMyostatin

Crude:              -0.288; 0.084

Adjusted:         -0.288; 0.087

DFM/FFM ratio -DInsulin

Crude:              0.358; 0.012

Adjusted:         0.335; 0.020

DFM/FFM ratio -DHOMA-IR

Crude:              0.524; 0.001

Adjusted:         0.487; 0.003

DSMM/weight - DHbA1c

Crude:              0.414; 0.004

Adjusted:         0.477; 0.001

DSMM/weight  - DHOMA-IR

Crude:              -0.335; 0.042

Adjusted:         -0.286; 0.089

DSMM/weight -DTotal-cholesterol

Crude:              -0.268; 0.046

Adjusted:         -0.326; 0.014

DASMM/BMI  -DHDL-cholesterol

Crude:              0.274; 0.043

Adjusted:         0.253; 0.065

DASMM/BMI  -Dnon-HDL-cholesterol

Crude:              -0.275; 0.042

Adjusted:         -0.316; 0.019

DASMM/BMI  -DCRP

Crude:              -0.352; 0.011

Adjusted:         -0.344; 0.015

D%Weight -D%HDL-cholesterol

Crude:              -0.338; 0.012

Adjusted:         -0.319; 0.019

D%Weight - D%Vitamin D

Crude:              -0.373; 0.006

Adjusted:         -0.389; 0.005

D%Weight - D%Myostatin

Crude:              -0.335; 0.033

Adjusted:         -0.321; 0.041

D%BMI - D%HDL-cholesterol

Crude:              -0.330; 0.014

Adjusted:         -0.310; 0.022

D%BMI - D% Triglycerides

Crude:              0.268; 0.046

Adjusted:        0.290; 0.034

D%BMI - D%Vitamin D

Crude:              -0.330; 0.016

Adjusted:         -0.344; 0.014

D%BMI - D%Myostatin

Crude:              -0.355; 0.023

Adjusted:         -0.342; 0.029

D%WC - D%IL-6

Crude:              -0.326; 0.019

Adjusted:        -0.325; 0.022

D%WC/Ht -  D%IL-6

Crude:              -0.326; 0.019

Adjusted:        -0.325; 0.022

D%Xc - D% Triglycerides

Crude:              0.284; 0.034

Adjusted:        0.284; 0.035

D%PhA/BMI - D%HbA1c

Crude:              0.397; 0.006

Adjusted:        0.427; 0.003

D%FM_% - D%HbA1c

Crude:             -0.324; 0.026

Adjusted:       -0.450; 0.003

D%FM_% - D%Insulin

Crude:              0.278; 0.053

Adjusted:        0.241; 0.104

D%FM_% -D%HOMA

Crude:              0.452; 0.005

Adjusted:        0.400; 0.020

D%FM_% -D%HDL-cholesterol

Crude:              -0.282; 0.037

Adjusted:         -0.256; 0.069

D%FM_kg - D%HbA1c

Crude:             -0.295; 0.044

Adjusted:       -0.391; 0.010

D%FM_kg -D%HOMA

Crude:              0.362; 0.028

Adjusted:        0.301; 0.077

D%FM_kg -D%HDL-cholesterol

Crude:              -0.288; 0.033

Adjusted:         -0.264; 0.057

D%FM_kg - D%CRP

Crude:              0.306; 0.029

Adjusted:        0.297; 0.039

D%FMI  -D%HOMA

Crude:              0.430; 0.008

Adjusted:        0.375; 0.026

D%FMI -D%HDL-cholesterol

Crude:              -0.273; 0.044

Adjusted:         -0.246; 0.078

D%FMI - D%CRP

Crude:              0.237; 0.094

Adjusted:        0.226; 0.120

D%Thigh SAT -D%Insulin

Crude:              0.362; 0.020

Adjusted:        0.346; 0.022

D%tASAT - D%HDL-cholesterol

Crude:              -0.460; 0.001

Adjusted:         -0.456; 0.001

D%tASAT - D% Triglycerides

Crude:              0.329; 0.023

Adjusted:         0.329; 0.024

D%sASAT - D%HDL-cholesterol

Crude:              -0.410; 0.004

Adjusted:         -0.408; 0.004

D%sASAT - D% Triglycerides

Crude:              0.368; 0.010

Adjusted:         0.368; 0.011

D%PreperitonealAT - D%Vitamin D

Crude:              -0.293; 0.051

Adjusted:         -0.297; 0.052

D%FFMI - D%HOMA

Crude:              -0.351; 0.033

Adjusted:        -0.343; 0.032

D%ASMMI -D%HDL-cholesterol

Crude:              -0.278; 0.040

Adjusted:         -0.263; 0.053

D%ASMMI -D%Vitamin D

Crude:              -0.339; 0.013

Adjusted:         -0.345; 0.013

D%RF-CSA - D%CRP

Crude:              -0.325; 0.036

Adjusted:         -0.316; 0.045

D%RF-CSA -D%Vitamin D

Crude:              0.309; 0.041

Adjusted:         0.317; 0.038

D%SMM/weight  - D%HbA1c

Crude:              0.460; 0.001

Adjusted:         0.513; <0.001

D%SMM/weight - D%HOMA

Crude:              -0.372; 0.023

Adjusted:         -0.327; 0.046

D%ASMM/BMI -D%HbA1c

Crude:              0.315; 0.031

Adjusted:         0.379; 0.011

D%ASMM/BMI -D%HDL-cholesterol

Crude:              0.284; 0.035

Adjusted:         0.265; 0.052

D%ASMM/BMI -D%non-HDL-cholesterol

Crude:              -0.264; 0.051

Adjusted:         -0.298; 0.028

D%ASMM/BMI -D%CRP

Crude:              -0.340; 0.015

Adjusted:         -0.334; 0.018

The variables were rank transformed to be included in the models. ASMM: Appendicular Skeletal Muscle Mass; ASMMI: Appendicular Skeletal Muscle Mass Index; AT: Adipose Tissue; BMI: Body Mass Index; CRP: C-reactive protein; ECW: Extracellular Water; FFM: Fat-Free Mass; FFMI: Fat-Free Mass Index; FM: Fat Mass; FMI: Fat Mass Index; HbA1c: glycosylated haemoglobin; HDL: High-density lipoprotein; HOMA-IR: Insulin Resistance Homeostatic Model Assessment; IL-6: Interleukin 6; PhA: Phase Angle; RF: rectus femoris; RF-CSA: rectus femoris cross-sectional area; SAT: subcutaneous adipose tissue; sASAT: superficial abdominal subcutaneous adipose tissue; SMM: Skeletal Muscle Mass; SMMI: Skeletal Muscle Mass  index; tASAT: total abdominal subcutaneous adipose tissue; TBW: Total Body Water; VF: Visceral Fat; WC: Waist Circumference; WHtR: Waist–to–Height Ratio; Xc: Reactance.

As the Spearman rho coefficient helped us exploratorily understand the associations between changes in body composition parameters and biochemical biomarkers, this new analysis allowed us to examine the average change in ranks of the dependent variables (biochemical biomarkers) for each one-rank increase in the studied body composition parameters (independent variables), adjusting for age.

When examining absolute differences, almost all relationships remained statistically significant and retained the same direction of association, as reflected by the standardized beta coefficients after adjustment. However, statistical significance was lost after adjustment in the following pairs: DASMMI (kg/m2) -DHDL-cholesterol (mg/dL), DSMM/weight (%) - DHOMA-IR, and DASMM/BMI (kg/kg*m2) -DHDL-cholesterol (mg/dL).

In the univariate analysis, associations between  DSMMI (kg/m2) - DTotal-cholesterol (mg/dL), DSMMI (kg/m2) - Dnon-HDL-cholesterol (mg/dL), and D RF thickness (cm) - DMyostatin (ng/mL) were not observed, suggesting that the correlations previously detected may represent more spurious associations.

When relative (%) differences where explored, age acted as a confounder in the following pairs: D%FM_% -D%HDL-cholesterol, D%FM_kg -D%HOMA, D%FM_kg -D%HDL-cholesterol, D%FMI -D%HDL-cholesterol, D%ASMMI -D%HDL-cholesterol, D%ASMM/BMI (kg/kg*m2) -D%non-HDL-cholesterol, and D%ASMM/BMI (kg/kg*m2) -D%HDL-cholesterol.

No significant associations were found in the univariate analysis for these pairs: D%FM_% - D%Insulin, D%FMI - D%CRP, and D%PreperitonealAT - D%Vitamin D.

We have added these paragraphs to the manuscript:

-Statistical section: “To account for confounders in the associations between changes in body composition parameters and biochemical biomarkers, we performed both crude and age-adjusted linear regression analyses using rank-transformed variables that showed statistically significant associations in the correlation analysis. Since the body composition parameters (independent variables) were interrelated and given the limited sample size, only age (rank-transformed) was included as a covariate.”

-Results section:

  1. Title: “Association” (instead of Correlations) between body composition changes and biomarkers evolution
  2. “The crude and age-adjusted linear regression models are presented in the supplementary table 3. Robust and biologically plausible associations were observed between body composition parameters and key metabolic biomarkers after adjustment for age. Increased FM, as measured by DFM (%), D%FM_% DFMI, DFM/FFM ratio, showed strong positive associations with DHOMA-IR and D%HOMA (adjusted β ranging from 0.400 to 0.487; p<0.025). Increased SAT, as measured by DtASAT, D%tASAT and D%sASAT demonstrated strong negative associations with DHDL-cholesterol and D%HDL-cholesterol (adjusted β ranging from 0.408 to 0.456; p<0.005). In addition, DsASAT was significantly and positively associated with DTriglycerides (adjusted β 0.405; p=0.005). However, some of the studied associations were no longer significant after adjusting for age.”

-Discussion section:

  1. “Our analyses revealed several robust and statistically significant associations between changes in body composition, classical anthropometric parameters and biochemical biomarkers, independent of age. Notably, increases in FM—expressed as percentage FM and FMI—were strongly and positively associated with markers of insulin resistance, including HOMA-IR. Conversely, SAT in thigh and abdominal regions showed consistent inverse relationships with HDL-cholesterol and positive associations with triglycerides, highlighting their impact on lipid metabolism.“
  2. “The results of both crude and age-adjusted linear regression models revealed several statistically significant associations between changes in body composition, classical anthropometric parameters, and biochemical biomarkers. However, some of these associations were no longer significant after adjusting for age. This attenuation of significance after adjustment suggests that age influences these associations and may partly explain the correlations observed in the crude analyses.”

2) Inclusion Criteria: More detail is required on recruitment, particularly regarding the impact of ongoing treatments on BC and biomarker evolution.

Thank you for this comment.

After informed consent, participants were consecutively recruited at the time of breast cancer diagnosis, prior to the initiation of any surgical, chemotherapeutic, hormonal, or radiotherapeutic treatment. Thus, the baseline assessment reflects the status before therapeutic intervention.

During the 12-month follow-up period, patients received various treatment regimens according to the clinical and biological characteristics of their tumors. These treatments included surgery (breast-conserving or mastectomy), chemotherapy, hormone therapy, and/or radiotherapy. Treatment data were collected to evaluate their impact on the progression of biomarkers and body composition in our cohort, considering possible interactions between treatment, baseline status, and clinical evolution. Given the sample size of this exploratory study, stratified analysis on this issue is beyond the scope of this manuscript (number of pairs for each treatment: radiotherapy/non-radiotherapy: 45/16; chemotherapy/non-chemotherapy: 14/47; hormonal therapy/non-hormonal therapy: 52/9), Nevertheless, this research will be conducted in the future, once the cohort is complete.

This information is summarized in

-Methods: “The study began in 2019, and recruitment is ongoing. Newly diagnosed breast cancer pa-tients (stages I–III) from the Breast Pathology Unit at Hospital Clínico San Carlos in Madrid were consecutively invited to participate. Participants were referred to the Nutrition Unit prior to the initiation of any treatment to undergo assessments of BC and muscle strength. Blood samples were collected, processed and stored the same day for biochemi-cal analysis. This assessment was scheduled to be repeated annually for up to five years.”

-Discussion: “Therefore, this sample size it has not allowed us to perform valid sub-analyses based on menopausal status or the treatment received, which have been identified as risk factors for deleterious changes in BC [34].”

3) Control Group: The absence of a control group significantly limits causal inferences.

We agree with you. We have now explicitly acknowledged this in the revised manuscript. However, preliminary studies have reported significant differences in body composition between breast cancer survivors and women without a cancer history, suggesting that cancer and its treatments may play a role in these changes. Although we were unable to include a matched control group in this initial phase, our ongoing study includes a long-term follow-up with an expanded cohort, which will allow for stratified analyses and better contextualization of these findings (Prevalence of sarcopenia in women with breast cancer. Nutrients 2022, 14, 1839. https://doi.org/10.3390/nu14091839 ).

This sentence was included in the limitations section:

The lack of a control group is another limitation of our study; however, preliminary reports suggest that cancer and its treatments may contribute to the observed changes in body composition.”

Morlino D, Marra M, Cioffi I, Santarpia L, De Placido P, Giuliano M, et al. Prevalence of Sarcopenia in Women with Breast Cancer. Nutrients. 2022; 14(9):1839. doi: 10.3390/nu14091839. PMID: 35565806; PMCID: PMC9099516.

4) Operator Variability: The US was performed by two individuals, which could introduce bias, yet inter-rater reliability is neither assessed nor mentioned.

Certainly, this is a significant concern for us. The researchers responsible for the US assessments were trained using the same standardized protocol (García-Almeida JM, et al. Nutritional ultrasound®: Conceptualisation, technical considerations and standardisation. Endocrinol Diabetes Nutr (Engl Ed). 2023; 70 Suppl 1:74–84. doi: 10.1016/j.endien.2022.11.010. PMID: 36935167). However, the obtained values are operator-dependent and may be subject to measurement bias.

We explored the intra- (3 measurements for each parameter) and inter-rater reliability: Inter-rater reliability (using RocTransvRF measured by three raters at the same time point): Intraclass Correlation Coefficient (ICC): 0.91. Intra-rater reliability (using RocTransvRF measured by a single rater over three time points): ICC: 0.94. These ICC values indicate excellent reliability for both inter- and intra-rater comparisons. The calculations were based on the commonly used two-way random effects, single measure, absolute agreement model.

We have added this paragraph to the discussion section (limitations):

“However, in an interim analysis, we observed excellent reliability for both inter-rater and intra-rater comparisons, with Intraclass Correlation Coefficients of 0.91 and 0.94, respectively.”

5) Statistical analysis: Several p-values are close to the significance threshold and should be interpreted cautiously. No adjustments for multiple comparisons are discussed, which increases the risk of Type I error. The rationale for choosing specific correlation coefficients and the implications of non-significant results are insufficiently discussed.

You are right in your concern. This is particularly relevant in the correlation analysis where 14 independent statistical tests were being conducted simultaneously. If Bonferroni correction is applied the value of p might be set at 0.004.

We are highlighted these values in the supplementary tables 1,2 and 3, and include a statement in the discussion section (limitations):

Given the multiple comparisons conducted simultaneously in the correlation and regression analyses, the reported significance levels should be interpreted with caution. Applying the Bonferroni correction, p values in the range of 0.004 to 0.05 may still carry a risk of Type I error. In Supplementary Tables 1, 2, and 3, the most robust p values have been highlighted.”

In the statistical methods, we have improved this sentence:

“Bivariate correlations between BC changes and biomarkers evolution (absolute -D- and relative -%- differences) were evaluated by Spearman's correlation coefficients after assessing the non-normality of the variables.

The implications of non-significant results are discussed in the limitations section, where the issue of limited statistical power is also addressed:

“The supplementary table 4 presents the results of post hoc power analyses for variables (body composition, strength and biochemical biomarkers) in which changes after one year of follow-up were not statistically significant. These analyses indicate that the non-significant findings may be attributable to insufficient statistical power, which ranged from 0.050 to 0.664. An increased sample size would be required to detect statis-tically significant differences. Moderate to large correlations (±0.35 and above) generally achieved acceptable power (>0.70), whereas weaker correlations (<0.30) were underpow-ered. Therefore, the lack of significance in correlations with low rho values in our study (approximately 0.10–0.20) may also be due to insufficient power. In contrast, significant correlations with moderate to large rho values were adequately powered.”

6) While changes in fat distribution and muscle parameters are well documented, the lack of significant changes in biomarkers undermines the strength of the manuscript’s key hypothesis

Thank you very much for paying attention to this limitation.

It is true that no statistically significant differences were observed in the evolution of the biomarkers; however, subtle changes that emerged during the first year of breast cancer treatment already allowed us to observe robust associations between changes in body composition and the progression of certain biomarkers—particularly those linking increased body fat with elevated HOMA values during follow-up, as well as those relating increased abdominal subcutaneous fat with decreased HDL cholesterol levels.

This discussion has been included in the revised manuscript (Discussion section):

“Our analyses revealed several robust and statistically significant associations between changes in body composition, classical anthropometric parameters and biochemical biomarkers, independent of age. Notably, increases in FM—expressed as percentage FM and FMI—were strongly and positively associated with markers of insulin resistance, including HOMA-IR. Conversely, SAT in thigh and abdominal regions showed consistent inverse relationships with HDL-cholesterol and positive asso-ciations with triglycerides, highlighting their impact on lipid metabolism.”

7) The rise in sarcopenic obesity prevalence (from 1.16% to 4.9%) is statistically minor and clinically questionable without stronger supporting data.

We sincerely appreciate your comment and understand your concern regarding the lack of significant changes in some biomarkers, as well as the interpretation of the increase in the prevalence of sarcopenic obesity.

Our primary objective was to describe the evolution of body composition and explore potential associations with biomarkers, rather than to establish causal relationships. If at any point our hypothesis was interpreted as aiming to demonstrate direct causality, we would like to clarify that this was not the purpose of the study. In fact, the available sample size significantly limits the ability to draw strong causal inferences.

Moreover, since this is a longitudinal study, we believe that longer-term follow-up will allow us to draw more robust conclusions than those currently possible in this early phase. We also consider that changes in body composition and biomarkers may occur gradually and subtly, which may explain the lack of statistically significant findings in some variables.

We expect that future studies with larger sample sizes and extended follow-up periods will help clarify these associations. Nonetheless, even if the lack of association is confirmed, this could still be valuable, as it may open new avenues for research and help to reformulate existing hypotheses.

For example, in the case of sarcopenic obesity diagnosis, current cut-off points may not be appropriate for younger populations, which is one of the main challenges when studying body composition in early adulthood. These types of limitations also highlight the importance of adapting diagnostic criteria to different age groups and population characteristics.

This paragraph has been modified in the text:

“SO has been proposed as a distinct BC phenotype, defined by the simultaneous presence of excess adiposity, low muscle mass, and reduced muscle strength. To date, only one study has analysed OS in breast cancer survivors based in data similar to the consensus definition proposed by ESPEN/EASO, reporting a prevalence of 16% [66]. This prevalence is considerably higher than that observed in our cohort (1.16–3.49% at baseline and at one-year follow-up, respectively). It should be noted that the referenced study in-cluded older patients, which may have contributed to a higher prevalence of sarcopenia.  In addition, the authors used different diagnostic criteria (SMM/weight <30.7%, FM >31.7% and HGS <20 kg) than those we applied in our analysis with a more conservative approach (SMM/weight <27.6%, FM >43% and HGS <16 kg). Nevertheless, our findings showed a slight increase in the prevalence of OS at one-year post-diagnosis, which may lend support to the previously discussed observations. However, this must be interpreted with caution given the limited number of women involved.

8) The discussion occasionally overstates conclusions from non-significant findings.

After extensive internal review of the manuscript, and with the valuable input from the external reviewers, we have improved the discussion section to provide a deeper examination of the main robust findings, while carefully avoiding overemphasis on weaker associations.

9) More attention should be paid to the influence of confounding variables (e.g., treatment regimens, physical activity, nutritional counseling).

Since the sample size is small, adjusting for other confounding variables is challenging, as previously discussed. In the ongoing study (MamaVida), all these factors will be considered, as data on dietary quality, physical activity, and other relevant factors are being collected.

This sentence has been modified in the limitation section:

“Therefore, this sample size it has not allowed us to perform valid sub-analyses based on menopausal status, exercise, dietary habits or the treatment received, which have been identified as risk factors for deleterious changes in BC [34].”

10) The link between muscle deconditioning and functional prognosis needs stronger justification, especially given that HGS remained unchanged.

Thank you for the recommendation, that is truly relevant. We have added the following paragraph to the discussion section:

In muscle deconditioning, the initial changes are often structural and may not be detectable with conventional assessment methods such HGS. One such example is myosteatosis which can precede functional decline. Typically, loss of muscle strength occurs at a later stage, once structural alterations are more established.”

11) The manuscript is readable but includes frequent grammatical errors, unclear sentence structures, and inconsistent terminology (e.g., "BC" for body composition is not universally recognized and should be standardized).

We have reviewed the syntax and replaced the acronym "BC" with "body composition," as the former could be confused with breast cancer.

Reviewer 4 Report

Comments and Suggestions for Authors

The study included merely 61 women, which is insufficient for multivariate correlations and subgroup analysis (e.g., menopausal state, therapy types). This limits the generalizability of findings and may result in underpowered statistical conclusions, especially for biomarker analyses with even fewer available data.
Recommendation: Provide a more specific justification for the sample size and recognize the statistical limitations more thoroughly in both the discussion and abstract.

This observational study lacks a non-breast cancer control group, so constraining the capacity to determine whether the observed alterations in body composition and biomarkers are from cancer diagnosis, treatment, or normal aging.
Recommendation: Provide a more robust justification for the absence of controls and examine how this affects the interpretation of the results.

The analysis reveals no statistically significant alterations in any of the assessed biomarkers after one year; however, the discussion seeks to establish some hypothetical connections to patterns. This undermines the validity of the conclusions.
Recommendation: Mitigate speculative interpretations or incorporate further statistical methodologies (e.g., adjustment for confounders or analysis of trends in responders versus non-responders) to substantiate biomarker significance.

The cohort underwent several treatments (chemotherapy, radiation, endocrine therapy), yet the text fails to stratify or account for their potential effects on muscle/fat alterations or biomarkers.
Recommendation: Incorporate subgroup or multivariate analysis that adjusts for therapy kind or time, or explicitly recognize this as a constraint.

Ultrasound and Bioelectrical Impedance Analysis were employed; however, the assessments were conducted by two distinct operators, potentially leading to inter-rater variability. The repeatability of these procedures is inadequately addressed.
Recommendation: Measure inter-operator reliability or incorporate ICCs (intra-class correlation coefficients) if accessible, and address this potential variability more explicitly.

Sarcopenic obesity (SO) is a significant result; nevertheless, the diagnostic thresholds employed are cautious and inconsistent with other data. The discourse fails to sufficiently examine the ramifications of SO in younger patients or to juxtapose it with other definitions.
Recommendation: Elucidate the rationale for diagnostic thresholds and examine their ramifications for clinical practice.

Certain muscle metrics diminished, whilst others, such as rectus femoris cross-sectional area and thickness, augmented. The argument regarding water retention or fat infiltration is conjectural in the absence of corroborating echogenicity or density evidence.
Recommendation: Include or at least address muscle echogenicity or MRI/CT data, if accessible, in future investigations; the existing discourse should explicitly categorize these as hypotheses.

The correlation heatmap, although instructive, is deficient in a suitable caption and accurate number labeling. The evaluation of several weak relationships may lead to exaggeration.
Recommendation: Enhance figure clarity, incorporate actual correlation coefficients in the figure or additional tables, and exercise caution when interpreting weak or marginal relationships.

The publication presents an inverse link between myostatin and RF thickness, as well as between Vitamin D and fat mass measurements; nonetheless, these results contradict certain previous investigations. This conflict has not been adequately explored.
Recommendation: Expound upon the divergence from existing literature and provide biological or methodological rationales.

The abstract presently contains superfluous methodological details and fails to clearly articulate the principal findings and their implications.
Recommendation: Condense the abstract to highlight principal findings, their clinical relevance, and limitations. Explicitly indicate that alterations in biomarkers were not statistically significant.

Author Response

Reviewer 4

Comments and Suggestions for Authors

1) The study included merely 61 women, which is insufficient for multivariate correlations and subgroup analysis (e.g., menopausal state, therapy types). This limits the generalizability of findings and may result in underpowered statistical conclusions, especially for biomarker analyses with even fewer available data.

Recommendation: Provide a more specific justification for the sample size and recognize the statistical limitations more thoroughly in both the discussion and abstract.

Thank you very much for helping us improve the manuscript. We have carefully considered all your suggestions.

Regarding sample size, we have extensively review the limitations of the sample size and the statistical power (see the supplementary table 4).

We have included in the abstract (last paragraph): “Further research is warranted to elucidate the long-term clinical implications of these findings and the external validity in larger cohorts.”

In the discussion:

“The small sample size is one of the main limitations of our study. From an initial re-cruitment of 138 participants, we selected the sample that completed the protocol after one year of follow-up, representing 44.2% of the cohort. Baseline characteris-tics—including age, hormonal status at diagnosis, tumour type, and comorbidities—were comparable between the total sample and this subset. Although such attrition may affect internal and external validity, the strongest changes and associations observed can be considered in future prospective studies, modelled with additional explanatory variables. The supplementary table 4 presents the results of post hoc power analyses for variables (body composition, strength and biochemical biomarkers) in which changes after one year of follow-up were not statistically significant. These analyses indicate that the non-significant findings may be attributable to insufficient statistical power, which ranged from 0.050 to 0.664. An increased sample size would be required to detect statis-tically significant differences. Moderate to large correlations (±0.35 and above) generally achieved acceptable power (>0.70), whereas weaker correlations (<0.30) were underpow-ered. Therefore, the lack of significance in correlations with low rho values in our study (approximately 0.10–0.20) may also be due to insufficient power. In contrast, significant correlations with moderate to large rho values were adequately powered.”

“Therefore, this sample size it has not allowed us to perform valid sub-analyses based on menopausal status, exercise, dietary habits or the treatment received, which have been identified as risk factors for deleterious changes in body composition.”

2) This observational study lacks a non-breast cancer control group, so constraining the capacity to determine whether the observed alterations in body composition and biomarkers are from cancer diagnosis, treatment, or normal aging.

Recommendation: Provide a more robust justification for the absence of controls and examine how this affects the interpretation of the results.

This is an important topic.

At the baseline measurement performed at the time of breast cancer diagnosis, surgical, systemic oncologic, and/or radiotherapeutic treatments had not yet been applied. Therefore, the body composition parameters obtained at this stage reflect the individual's physiological baseline status, without the influence of therapeutic interventions. However, it is possible that some biochemical parameters were already altered as a consequence of the presence of the tumor and, thus, may not fully represent a "normal" baseline state. This study focused on characterizing longitudinal changes in body composition and biomarkers in a cohort of breast cancer survivors during the first year following diagnosis. Given the absence of a cancer-free control group, comparisons were made with normative data available in nutrition guidelines to provide a reference perspective, although these data are not specific to oncology patients nor do they fully distinguish between the effects of cancer and normal aging. We acknowledge this methodological limitation and address it in the manuscript’s discussion section, evaluating its impact on the interpretation and generalization of the findings:

“The lack of a control group is another limitation of our study; however, preliminary re-ports suggest that cancer and its treatments may contribute to the observed changes in body composition [71].”

3) The analysis reveals no statistically significant alterations in any of the assessed biomarkers after one year; however, the discussion seeks to establish some hypothetical connections to patterns. This undermines the validity of the conclusions.

Recommendation: Mitigate speculative interpretations or incorporate further statistical methodologies (e.g., adjustment for confounders or analysis of trends in responders versus non-responders) to substantiate biomarker significance.

We agree with these considerations and have made the corresponding revisions accordingly.

-We have thoroughly reviewed the discussion section to appropriately moderate the emphasis placed on the significance of the findings and to curtail speculative interpretations.

-We have assessed the relationship between body composition and biochemical biomarkers evolution in linear regression models adjusted by age (see Supplementary table 3, statistics section, Results and Discussion).

4 ) The cohort underwent several treatments (chemotherapy, radiation, endocrine therapy), yet the text fails to stratify or account for their potential effects on muscle/fat alterations or biomarkers.

Recommendation: Incorporate subgroup or multivariate analysis that adjusts for therapy kind or time, or explicitly recognize this as a constraint.

Thank you for this suggestion.

We have explained the limitations in the discussion:

“Therefore, this sample size it has not allowed us to perform valid sub-analyses based on menopausal status, exercise, dietary habits or the treatment received, which have been identified as risk factors for deleterious changes in body compositionBC [34]”

This statement is based on this previous commentary in this cover letter:

“As the main project focuses on the evolution of adiposity over time following breast cancer diagnosis, the sample size calculation for this exploratory analysis was based on detecting changes in the most clinically accessible adiposity marker, %FM estimated by BIA. Accordingly, a sample size of 44 paired observations would be required to reliably detect an effect size of δ ≥ 0.5 with a statistical power greater than 0.9, assuming a two-sided test and a maximum Type I error rate of α = 0.05. For US comparisons, the sample size estimate was more conservative due to the lack of established data on minimally clinically relevant changes. In this context, 52 paired observations would be needed to detect an effect size of δ ≥ 0.4 with a power greater than 0.8, also assuming a two-sided test and α = 0.05.”). The potential advantages conducting stratified analysis may be outweighed by the drawbacks or limitations, given the limited number of pairs available in each subgroup (premenopausal/postmenopausal: 16/45; radiotherapy/non-radiotherapy: 45/16; chemotherapy/non-chemotherapy: 14/47; hormonal therapy/non-hormonal therapy: 52/9; luminal A and B/other types: 41/20).

5) Ultrasound and Bioelectrical Impedance Analysis were employed; however, the assessments were conducted by two distinct operators, potentially leading to inter-rater variability. The repeatability of these procedures is inadequately addressed.

Recommendation: Measure inter-operator reliability or incorporate ICCs (intra-class correlation coefficients) if accessible, and address this potential variability more explicitly.

As discussed previously:

Certainly, this is a significant concern for us. The researchers responsible for the US assessments were trained using the same standardized protocol (García-Almeida JM, et al. Nutritional ultrasound®: Conceptualisation, technical considerations and standardisation. Endocrinol Diabetes Nutr (Engl Ed). 2023; 70 Suppl 1:74–84. doi: 10.1016/j.endien.2022.11.010. PMID: 36935167). However, the obtained values are operator-dependent and may be subject to measurement bias.

We explored the intra- (3 measurements for each parameter) and inter-rater reliability: Inter-rater reliability (using RocTransvRF measured by three raters at the same time point): Intraclass Correlation Coefficient (ICC): 0.91. Intra-rater reliability (using RocTransvRF measured by a single rater over three time points): ICC: 0.94. These ICC values indicate excellent reliability for both inter- and intra-rater comparisons. The calculations were based on the commonly used two-way random effects, single measure, absolute agreement model.

We have added this paragraph to the discussion section (limitations):

“However, in an interim analysis, we observed excellent reliability for both inter-rater and intra-rater comparisons, with Intraclass Correlation Coefficients of 0.91 and 0.94, respectively.”

6 ) Sarcopenic obesity (SO) is a significant result; nevertheless, the diagnostic thresholds employed are cautious and inconsistent with other data. The discourse fails to sufficiently examine the ramifications of SO in younger patients or to juxtapose it with other definitions.

Recommendation: Elucidate the rationale for diagnostic thresholds and examine their ramifications for clinical practice.

We have addressed this topic with interest, though not without concern. As previously discussed in this cover letter. The following paragraph has been added to the Discussion section:

Current cut-off points for sarcopenic obesity diagnosis may not be appropriate for younger populations, which is one of the main challenges when studying body composition in early adulthood. These types of limitations also highlight the importance of adapting diagnostic criteria to different age groups and population characteristics.”

7) Certain muscle metrics diminished, whilst others, such as rectus femoris cross-sectional area and thickness, augmented. The argument regarding water retention or fat infiltration is conjectural in the absence of corroborating echogenicity or density evidence.

Recommendation: Include or at least address muscle echogenicity or MRI/CT data, if accessible, in future investigations; the existing discourse should explicitly categorize these as hypotheses.

Thank you for this recommendation regarding future research. We are very interested in studying muscle echogenicity in these patients, as it serves as an index of muscle quality that may be more relevant than muscle size. Since simple visual observations are subjective and grayscale interpretation can be challenging, we look forward to acquiring AI-based software capable of analyzing muscle echogenicity. This advancement will enable us to evaluate changes in muscle quality more accurately and will strengthen the validity of our complementary measurements.

This sentence has been added to the future research planning:

AI-based software capable of analyzing muscle echogenicity will enable us to evaluate changes in muscle quality more accurately and will strengthen the validity of our complementary measurements.

8) The correlation heatmap, although instructive, is deficient in a suitable caption and accurate number labeling. The evaluation of several weak relationships may lead to exaggeration.

Recommendation: Enhance figure clarity, incorporate actual correlation coefficients in the figure or additional tables, and exercise caution when interpreting weak or marginal relationships.

We agree with you that the heat map should be interpreted with caution. As previously explained:

We understand that the figure may be mistakenly interpreted, leading readers to conclude that some correlations are significant when they are not. This was not the authors' intention. In an exploratory study such as ours, which involves the analysis of multiple variables, the heatmap was intended as a tool for visually identifying relevant associations and summarizing the data in an interpretable manner. The complete data, including all Spearman’s rho correlation coefficients and their corresponding p-values, are provided in Supplementary Tables 1 and 2.

Nevertheless, if the reviewer and editor consider that removing the figure would improve the manuscript, we will accept that decision.

Additionally, we have added the following note as a footnote to the figure:

“Spearman’s rho correlation coefficients and their p-values are described in Supplementary Tables 1 and 2.”

9) The publication presents an inverse link between myostatin and RF thickness, as well as between Vitamin D and fat mass measurements; nonetheless, these results contradict certain previous investigations. This conflict has not been adequately explored.

Recommendation: Expound upon the divergence from existing literature and provide biological or methodological rationales.

Thank you for your attention to this important topic, which we plan to study in greater depth in future research.

Our findings indicating an inverse relationship between circulating myostatin levels and rectus femoris muscle thickness, as well as between Vitamin D and fat mass measurements, align with biological mechanisms whereby myostatin acts as a negative regulator of muscle growth and Vitamin D status is often inversely related to adiposity. However, these associations contrast with some prior studies reporting inconsistent or null correlations, which may be attributable to differences in assay sensitivity, participant characteristics, and study design. In breast cancer survivors, these discrepancies could be further influenced by the complex interplay of cancer pathology, treatment effects, and systemic metabolic alterations that differ from general populations. Moreover, circulating biomarker levels may not fully reflect localized tissue activity or fat distribution nuances. Methodological factors such as measurement modality and sample size could also contribute to divergent results. Consequently, while our data suggest biologically plausible links, these findings warrant cautious interpretation and highlight the need for longitudinal studies with larger cohorts and integrated biochemical and imaging approaches to elucidate the mechanisms underpinning muscle and fat changes in breast cancer survivors.

10) The abstract presently contains superfluous methodological details and fails to clearly articulate the principal findings and their implications.

Recommendation: Condense the abstract to highlight principal findings, their clinical relevance, and limitations. Explicitly indicate that alterations in biomarkers were not statistically significant.

Thank you for this valuable suggestion. The abstract has been changed:

Abstract: Background/Objectives: After diagnosis, it is common for women with breast cancer to gain weight, which is associated with worse clinical outcomes. However, tradi-tional measures such as body weight, BMI, and waist circumference do not detect key changes in body composition, such as fat redistribution or muscle loss. The objective of this exploratory study was to assess the evolution of body composition and muscle strength after one year of treatment, and their relationship with metabolic biomarkers. Methods: Prospective observational study in newly diagnosed breast cancer patients. Body composition was assessed using bioelectrical impedance analysis (BIA) and ultra-sound (US); muscle strength was measured by handgrip dynamometry. Biomarkers ana-lyzed included glucose, insulin, Homeostatic Model Assessment of Insulin Resistance (HOMA-IR), glycosylated hemoglobin (HbA1c), total cholesterol (and its fractions), tri-glycerides, C-reactive protein (CRP), 6-interleukine (IL-6), vitamin D, myostatin and Fi-broblast Growth Factor 21 (FGF-21). Results: Sixty-one women (mean age 58 years) were included. After one year, fat mass and related parameters significantly increased, while skeletal muscle mass and muscle strength decreased. Sarcopenic obesity prevalence rose from 1.16% to 4.9%. No significant changes were found in biomarkers, but positive cor-relations were observed between fat parameters and insulin, HOMA-IR, and triglycer-ides, and negative correlations with HDL-cholesterol. Conclusions: BIA and US can de-tect unfavourable changes in body composition that are not reflected in conventional measurements. One-year post-diagnosis, women showed increased fat accumulation, muscle loss, and reduced strength, even without significant metabolic biomarker changes. Further research is warranted to elucidate the long-term clinical implications of these findings and the external validity in larger cohorts.”

Round 2

Reviewer 1 Report

Comments and Suggestions for Authors

.

Reviewer 2 Report

Comments and Suggestions for Authors

Accept in present form

Reviewer 3 Report

Comments and Suggestions for Authors

Satisfied

Reviewer 4 Report

Comments and Suggestions for Authors

The author improve well.